# DeepSADR: Deep Transfer Learning with Subsequence Interaction and Adaptive Readout for Cancer Drug Response Prediction

**Yuanpeng Zhang**[1], **Zhijian Huang**[1], **Ziyu Fan**[1], **Siyuan Shen**[1], **Yahan Li**[1], **Shangqian Wu**[1], **Min Wu**[2,*], **Lei Deng**[1,*]
[1]School of Computer Science and Engineering, Central South University, ChangSha, China
[2]Institute for Infocomm Research (I[2]R), A*STAR, Singapore
{yuanpengzhang,leideng}@csu.edu.cn, wumin@a-star.edu.sg
*Corresponding author

## Abstract

Cancer treatment efficacy exhibits high inter-patient heterogeneity due to genomic variations. While large-scale in vitro drug response data from cancer cell lines exist, predicting patient drug responses remains challenging due to genomic distribution shifts and the scarcity of clinical response data. Existing transfer learning methods primarily align global genomic features between cell lines and patients. However, they often ignore two critical aspects. First, drug response depends on specific drug substructures and genomic function subsequences. Second, drug response mechanisms differ in vitro and in vivo settings due to factors such as the immune system and tumor microenvironment. To address these limitations, we propose DeepSADR, a novel deep transfer learning framework for enhanced drug response prediction based on subsequence interaction and adaptive readout. In particular, DeepSADR models drug responses as interpretable bipartite interaction graphs between drug substructures and genomic function subsequences. Subsequently, a supervised graph autoencoder was designed to capture latent interactions between drugs and gene subsequences within these interaction graphs. In addition, DeepSADR treats the drug response process as a transferable domain. A Set Transformer-based adaptive readout (AR) function learns domain-invariant response representations, enabling effective knowledge transfer from abundant cell line data to scarce patient data. Extensive experiments on clinical patient cohorts demonstrate that DeepSADR significantly outperforms state-of-the-art methods, and ablation experiments have validated the effectiveness of each module.

## 1 Introduction

Cancer is the leading cause of morbidity and mortality worldwide (Fan et al., 2019). Genomic heterogeneity drives strong variability in drug response across patients, necessitating precise predictive models for personalized therapy. To aid in treatment development, large-scale studies have been conducted globally, such as the Cancer Genome Atlas (TCGA) database (Hutter & Zenklusen, 2018) to compile high-dimensional genomic information from cancer patients. However, patient drug response data in current databases are extremely scarce, primarily due to limited patient cohorts and the fact that each patient typically receives only a few drugs. For example, TCGA contains only 500 patient drug response cases (Sharifi-Noghabi et al., 2020). To overcome this limitation, researchers often rely on pre-clinical datasets, especially cancer cell lines. Cell lines are derived from patient tumors and cloned to maintain stable genomic profiles. These cloned cells can be exposed to many different drugs. This allows researchers to collect drug response data across multiple drugs within the same cell line. Such data is highly valuable as it cannot be collected directly from patients due to the risks of administering multiple drugs concurrently. Although currently constrained to roughly 1,000 cell lines and a limited number of drugs, it nonetheless provides an essential foundation for developing personalized drug response models based on genomic information.

Currently, many researchers have proposed predictive models based on drug-cancer cell line response data, such as DeepCovDR (Huang et al., 2023), GraphCDR (Liu et al., 2022) and Deep-ExpDR (Zhang et al., 2025), which are deep learning models. These deep learning models have shown strong performance in predicting drug responses in cancer cell lines. However, studies have shown that such models often fail to accurately predict drug efficacy in patients (Seyhan, 2019b). One major reason is the distributional gap between cell lines and patient data. Genomic profiles of cell lines ($\mathcal{G}^c$) are more homogeneous than those of patients ($\mathcal{G}^p$). This results in distinct distributions of genomic information ($P(\mathcal{G}^c) \neq P(\mathcal{G}^p)$). As shown in the t-SNE visualization in Appendix A.10 (Figure 4), the distributions of cell lines and patients differ significantly. As such, they can be regarded as coming from different domains (see Appendix A.2).

To address these challenges, researchers have developed various drug response models based on domain adaptation and transfer learning. These methods attempt to bridge this distributional gap typically learn domain-invariant feature representations shared between cell lines (source domain) and patients (target domain). Despite their progress, existing approaches still face important limitations. First, they often ignore certain important functional fragments. Treating drugs and genes as monolithic entities overlooks key drivers, such as drug pharmacophores and enriched genomic pathways (e.g., apoptosis). For instance, variations in tumor suppressor genes (TSGs) are major contributors to paclitaxel resistance across cancers (Xu et al., 2016). Similarly, the benzodiazepine scaffold in the anticancer drug Devazepide is active against opioid receptors and other protein targets (Marsters Jr et al., 1994). Second, many methods focus solely on the distribution differences in genomic characteristics between cell lines and patients, yet overlook a critical biological fact: there are systemic differences in the mechanisms by which drugs respond in vitro cell lines versus in vivo patient environments. These differences stem from factors such as the tumor microenvironment, the immune system, and systemic physiological factors—elements that cannot be fully replicated in cell line models (Seyhan, 2019a).

In this paper, we propose DeepSADR, a transfer learning model for drug response prediction from cell lines to patients, built on subsequence interaction and adaptive readout. DeepSADR adopts pretraining and fine-tuning strategy. It constructs subsequence interaction graphs to capture associations between drug and gene subsequences, which improves both performance and interpretability. To enable effective knowledge transfer in the drug response domain, we introduce an adaptive readout function that learns domain-invariant features, thus enhancing the model's predictive performance on patient data. The contributions of this work are summarized as follows.

- We model drug responses as bipartite interaction graphs between drug subsequences and gene subsequences. A supervised graph autoencoder is then used to capture their complex associations in an interpretable way.

- We propose an adaptive readout based on the ensemble transformer architecture that effectively aggregates node features from the subsequence interaction graph into a global drug response representation. During fine-tuning, we also incorporate pre-trained drug response embeddings to enhance features and learn domain-invariant representations. This design overcomes the limitations of standard pooling functions in graph transfer learning.

- We treat the entire drug response biological process with cell lines/patients as a distinct domain for transfer learning, moving beyond simple genomic feature alignment to better address inherent biological differences.

- Extensive experiments show that DeepSADR significantly improves drug response prediction performance (AUC/AUPR) on scarce clinical patient data and provides interpretability through interaction visualization.

## 2 RELATED WORK

### 2.1 DRUG RESPONSE PREDICTION

Currently, drug response prediction (DRP) models for patients mainly focus on transfer learning between cell lines (source domain) and patients (target domain). These methods can be classified as inductive, transductive, or unsupervised, depending on whether labeled patient data is used. Inductive methods include drug2tme (Zhai & Liu, 2024), PREDICT-AI (Jayagopal et al., 2024) and

GANDALF (Jayagopal et al., 2025b), which utilize both labeled cell lines and patient samples to capture differences in label distributions between the two domains through Deep learning. However, this approach heavily relies on labeled patient data, which is often difficult, expensive, and scarce in clinical practice. A few methods employ unsupervised approaches, such as CODE-AE (He et al., 2022), which uses unlabeled cell lines and patient data for pre-training. Transduction-based methods include TUGDA (Peres da Silva et al., 2021), WISER (Shubham et al., 2024) and PANCDR (Kim et al., 2024), which utilize labeled cell lines and unlabeled patient samples. Inductive and transductive methods are currently the most widely used approaches. These methods primarily aim to learn shared representation spaces across domains, thereby mitigating the distributional differences between cell lines and patient data. While shared representations can capture similarities across different domains and improve model predictive performance to some extent, these methods do not adequately account for gene data fragments and drug substructures that play a crucial role in drug responses, nor do they sufficiently consider the distributional differences in the biological mechanisms underlying drug responses(Drug-patient responses are influenced by numerous biological factors). — which is critical for predicting drug responses in patients.

## 2.2 SUBSEQUENCE SEGMENTATION

In recent years, subsequence segmentation methods have been applied in many fields of bioinformatics and machine learning. For example, explainable sub-structure fingerprinting (ESFP) is a substructure-based fingerprint representation method (Huang et al., 2019) that constructs molecular fingerprints by identifying and quantifying specific sub-structures in molecules, which is helpful for drug and protein research. Conventional DRP models typically extract features from drug and gene transcription information separately and then combine them as features for drug responses. This approach is overly simplistic and fails to consider the drug/gene subsequences that play a crucial role in drug responses. This paper innovatively combines gene transcription subsequences with drug SMILES subsequences, transforming the entire drug response process into a subsequence interaction graph. Drug/Gene subsequences serve as nodes in this graph, and the features of the entire subsequence interaction graph are used as features for drug responses.

## 2.3 ADAPTIVE READOUT

The readout function is a critical component in Graph Neural Network(GNN) for processing graph-level tasks, as it transforms node representations into graph representations. Common readout functions include simple ones such as summation, averaging, and maximum values. We transform the drug response process into a subsequence interactions graph, so selecting an appropriate readout function is critical to the final performance of the transfer learning model. The Set Transformer is a Transformer model designed for set-based data, which handles unordered inputs through permutation invariance and is suitable for tasks where the order of elements is irrelevant. Therefore, we designed an adaptive readout function based on the Set Transformer concept. We incorporated the embedded vectors output by the pre-trained model function into the model fine-tuning stage to enhance features, thereby overcoming the limitations of traditional readout functions in transfer learning for patient drug response data.

## 3 PROPOSED METHOD

### 3.1 PROBLEM DEFINITION

Suppose there are $N_c$ labeled drug-cell line genomic profiles($\mathcal{G}^c$) response data and $N_p$ labeled drug-patient genomic profiles($\mathcal{G}^p$) response data. In general, $N_c >> N_p$. Let $D = \{d_1, d_2, d_3, ..., d_N\}$ be the set of N drugs with labeled drug responses, where $\mathcal{Y}_c(d_i, g_j^c)$ denotes the corresponding response of a drug $d_i$ to a cell lines genomic profile $g_j^c \in \mathcal{G}_c$, and $\mathcal{Y}_p(d_i, g_k^p)$ be the drug response for patients $g_k^p \in \mathcal{G}_p$. Note that $\mathcal{Y}_c(d_i, g_j^c)/\mathcal{Y}_p(d_i, g_k^p) \in \{1, 0\}$ where 1 indicates a positive response, 0 indicates a negative response. The primary goal of our work is to leverage large-scale labeled drug response data from cell lines (source domain) via transfer learning, to significantly improve the prediction of drug response in patients (target domain) who have small sample sizes.

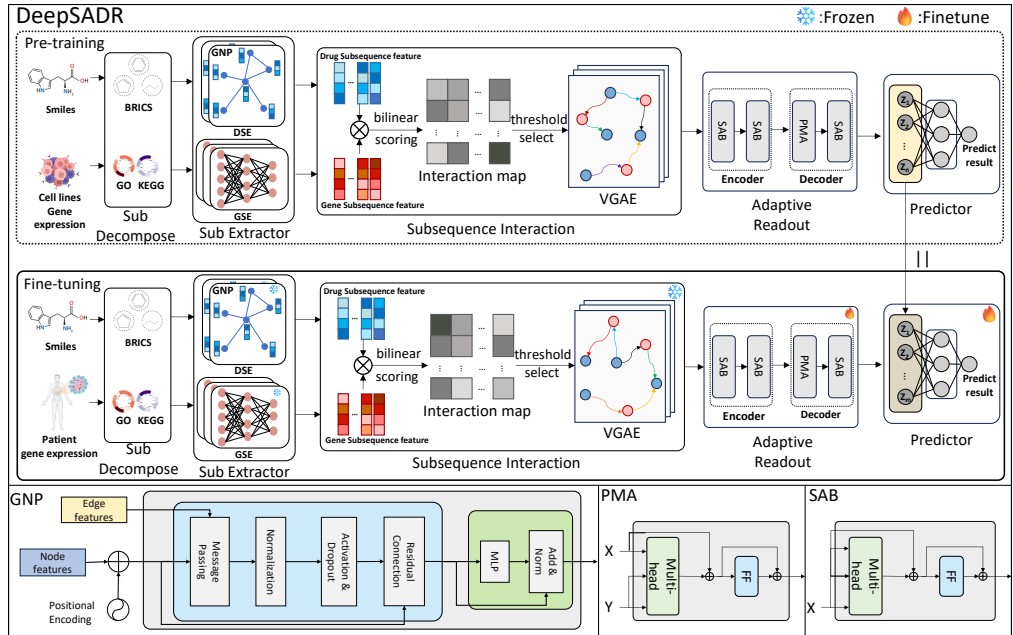

Figure 1: Overview of the DeepSADR. (A) shows the model architecture, which is divided into two stages: pre-training and fine-tuning. The model consists of five components: Sub Decompose, Sub Extractor, Subsequence Interaction, Adaptive Readout, and Predictor. During fine-tuning, only the adaptive readout and predictor modules undergo training. The latent space embeddings generated during pre-training serve as feature inputs to the fine-tuning stage, enabling the model to learn domain-invariant drug response features and achieve efficient knowledge transfer.(B) shows the framework of the GNP module. (C) shows the framework of the PMA module. (D) shows the framework of the SAB module.

## 3.2 METHOD OVERVIEW

DeepSADR is a deep learning model used to improve the predictive performance of drug responses in patients (DeepSADR follows an inductive transfer learning framework, where we fine-tune the model using labeled cell line data (source) and a small subset of labeled patient data (target), similar to the GANDALF(Jayagopal et al., 2025b) approaches). Its overall architecture is shown in Figure 1. The main process of the model is divided into two stages: pre-training and fine-tuning, and consists of the following four key modules:

- **Sub Decompose**: Biologically meaningful decomposition of drug molecules and genomic profiles.

- **Sub Extractor**: Feature extraction of drug and gene subsequences.

- **Subsequence Interaction**: Construct an interaction graph between drug subsequences and gene subsequences. Use a supervised graph autoencoder to learn latent representations in order to capture the complex relationships and interaction strengths between them.

- **Adaptive Readout**: Through a readout function based on Set Transformer, subsequence interaction graphs are integrated into a global drug response representation. This adaptive readout mechanism is crucial for learning transferable features across different domains (cell lines and patients).

## 3.3 SUB DECOMPOSE

Extensive research indicates that within the DRP framework, the drug response process is not only related to drug structure and gene profiles information but may also be highly correlated with more granular 'subcomponents', such as the substructures of drug molecules and the genetic sub-fragment characteristics of cancer cells. However, most current drug response prediction models perform

feature extraction on drugs/genes as a whole, which to some extent overlooks certain important subcomponents and makes it more difficult for the model to explain how drugs treat patients. Based on this, this paper performs 'subcomponent' decomposition on drug and genomic data.

**Drug Smiles Decompose.** For the decomposition of drug SMILES sequences, we utilize the core method (**BRICS**) for molecular decomposition in the RDKit library (Degen et al., 2008). The formula is as follows:

$$\textbf{BRICS}(\mathcal{S}(d_i)) \longrightarrow [\mathcal{S}_{sub}^1, \mathcal{S}_{sub}^2, ..., \mathcal{S}_{sub}^n], \tag{1}$$

where $\mathcal{S}(d_i)$ denotes the Smiles sequence of drug $d_i \in D$, $\mathcal{S}_{sub}$ denotes the subsequence of the drug Smiles obtained through BRICS decomposition, and the superscript denotes the corresponding index, $n$ denotes the total number of drug subsequences.

**Genomic Profiles Decompose.** We used the gseapy library (Fang et al., 2022) to perform functional enrichment analysis on the gene lists of cell lines/patients to identify significantly enriched biological processes (such as "apoptosis" and "DNA repair") and reveal the biological functions that these genes may share. For example, if the input genes are cancer differentially expressed genes, the enrichment analysis may find that they are significantly enriched in the "cell cycle regulation" or "immune response" pathways, suggesting that these processes are related to cancer. Based on the gene pathway results, we designed a gene functional clustering algorithm to assign these genes to generalized functional pathway clusters. The formula is as follows:

$$\textbf{ENRICH}(g_j^c/g_j^p) \longrightarrow [\mathcal{G}_{sub}^1, \mathcal{G}_{sub}^2, ..., \mathcal{G}_{sub}^m], \tag{2}$$

where $g_j^c/g_j^p$ denotes the genomic profile of cell lines/patients, **ENRICH** denotes gene function clustering algorithm (For more details, see Appendix A.20.), $\mathcal{G}_{sub}$ denotes the subsequence of the genomic profiles, and the superscript denotes the corresponding index, $m$ denotes the total number of genomic profiles subsequences.

## 3.4 SUB EXTRACTOR

**Drug Sub Extractor.** Traditional methods for processing drug SMILES sequences generally use GNN (Scarselli et al., 2009), which represent molecules as graphs (atoms = nodes, chemical bonds = edges). Although this method has achieved relatively good results, its message passing mechanism is limited by local neighborhood aggregation, making it difficult to capture non-bond interactions and model long-range effects across molecules. To address the issue of limited message passing in GNNs and effectively extract features from drug subsequences, based on the research of (Luo et al., 2025), we integrated six techniques into the classic GNN: edge feature integration, normalization, Dropout, residual connections, feedforward networks (FFN), and positional encoding. These techniques were combined to form a $GNP$ feature encoder for feature encoding of drug subsequences. $GNP$ is shown in Figure 1, and its specific formula is as follows:

$$Sub_i^d = DSE(BRICS(\mathcal{S}(d_i))), \tag{3}$$

$$DSE(\mathcal{S}_{sub}^1, ..., \mathcal{S}_{sub}^n) = \{GNP(\mathcal{S}_{sub}^1), ..., GNP(\mathcal{S}_{sub}^n)\}, \tag{4}$$

where $DSE$ represents 'Drug Sub Extractor', $Sub_i^d \in \mathbb{R}^{(n \times e_d)}$ denotes the all drug subsequence features of the i-th drug. More details of $GNP$ in Appendix A.3.

**Gene Sub Extractor.** For gene subsequences, we use m fully connected layers for preliminary feature extraction, as shown in the following formula:

$$Sub_j^c/Sub_j^p = GSE(enrichr(g_j^c/g_j^p)), \tag{5}$$

$$GSE(\mathcal{G}_{sub}^1, ..., \mathcal{G}_{sub}^m) = \{f_1(\mathcal{G}_{sub}^1), ..., f_m(\mathcal{G}_{sub}^m)\}, \tag{6}$$

where $GSE$ represents 'Gene Sub Extractor', $Sub_j^c/Sub_j^p \in \mathbb{R}^{(m \times e_g)}$ denotes the all gene subsequence features of the j-th cell line/patient and $ge$ the dimension of gene subsequence features, $f$ denotes fully connected layer.

## 3.5 SUBSEQUENCE INTERACTION

In order to explore the potential associations between drug subsequences and gene subsequences and enhance the interpretability of the model, we constructed a subsequence interaction graph using

the subsequence features of drugs and genes, and then extracted features from the subsequence interaction graph using a supervised graph autoencoder (Kipf & Welling, 2016a).

**Construction of subsequence interactions graph.** We design an interaction function $\psi$ with a simple bilinear score to measure the interaction between each subsequence in the drug and each subsequence in the gene. The specific formula is as follows:

$$\mathcal{R} = \psi(Sub_i^d, Sub_j^g), \tag{7}$$

$$\psi(\hat{d}, \hat{g}) = \sigma(\hat{d}w\hat{g}^\top), \tag{8}$$

where $w \in \mathbb{R}^{e_d \times e_g}$ represents a trainable parameter matrix. $Sub_j^g$ is a variable representing the cell line subsequence features $Sub_j^c$ during the pre-training phase and the patient subsequence features $Sub_j^p$ during the fine-tuning stage. $\hat{d}$ and $\hat{g}$ are the subsequence features of the drug and gene, respectively. $\sigma$ denotes a sigmoid activation function and the function $\psi$ outputs $\mathcal{R} \in R^{n \times m}$ is a two-dimensional scalar matrix (interactions score), each value in the matrix ranges from [0,1] and represents the strength of interaction between each subsequence. Therefore, we can regard each drug response process as a subsequence interaction map. If a pair of subsequence significantly contributes to the prediction result, they will be updated during training and obtain a higher score at the corresponding position in the graph. The trained graph can provide key insights into which subsequences influence drug response outcomes, thereby enhancing the model's interpretability.

**Supervised graph autoencoder.** Since we view the drug response process as a subsequence interaction graph, how can we adequately consider the relations between subsequences in the interaction graph to obtain high-quality drug response features that are conducive to transfer learning? Inspired by the powerful capabilities of graph convolutional networks, we propose formalizing the subsequence interaction graph and subsequence features as a bipartite graph structure $G(\mathbf{A}, \mathbf{X})$, where $\mathbf{X} = \{Sub_i^d, Sub_j^c / Sub_j^p\}$ denotes the feature set corresponding to the two entities (drug subsequence features and gene subsequence features), $\mathbf{A} \in \mathbb{R}^{(n+m) \times (n+m)}$ is an adjacency matrix obtained by threshold selection from $\mathcal{R}$. The specific process is as follows:

$$\mathbf{A} = \begin{pmatrix} 0_{n \times n} & \hat{\mathcal{R}} \\ \hat{\mathcal{R}}^\top & 0_{m \times m} \end{pmatrix}, \tag{9}$$

$$\hat{\mathcal{R}}[i][j] = \begin{cases} \mathcal{R}[i][j] & \mathcal{R}[i][j] \geq t; \\ 0 & \mathcal{R}[i][j] < t, \end{cases} \tag{10}$$

where $t$ represents the threshold, which is a selectable parameter. Since the subsequence interactions graph we constructed is a complete graph, in reality, some subsequences are independent of each other and may have no association. Therefore, we introduce a threshold selection operation to remove some association edges with smaller weights from the subsequence interaction graph (we can regard this edges as noise), reducing unnecessary interference and improving the efficiency and performance of the model.

We then use the encoder in SGAE to extract features from the interaction graph, thereby capturing and aggregating the correlations between all interactions. The specific process is as follows:

$$\mathbf{Z} = SGAE(\mathbf{X}, \mathbf{A}). \tag{11}$$

More details of $SGAE$ can be found in Appendix A.4.

### 3.6 ADAPTIVE READOUT

Unlike methods focusing solely on genomic distribution differences between source (cell lines) and target (patients) domains, we conceptualize the entire drug response process as a distinct domain. We then employ transfer learning (Pan & Yang, 2010) to mitigate distribution shifts specifically within this drug response domain between source and target. Therefore, how to readout drug response representations from subsequence interactions graph has a significant impact on transfer learning. Traditional graph readout methods use fixed pooling functions (such as sum/mean/max) to aggregate node embeddings into graph embeddings. This readout lacks flexibility and is not conducive to transfer learning. In this study, we design an adaptive readout function based on the 'Set Transformer' (Lee et al., 2019) to aggregate node embeddings into graph embeddings. This readout

function learns domain-invariant representations by aggregating node embeddings in a permutation-invariant manner, capturing complex interactions between sub-sequences. It's trainable parameters enable fine-tuning of the readout layer as a feasible and efficient transfer strategy. During fine-tuning, we freeze other modules of the pre-trained model while updating only the Adaptive Readout (AR) and predictor, allowing the model to adapt to patient-specific biological mechanisms (e.g., immune responses, tumor microenvironments) without overfitting. The readout function is as follows:

$$\mathcal{Z} = AR(\mathbf{Z}) = \frac{1}{K} \sum_{k=1}^{K} \left[ Decoder \left( Encoder(\mathbf{Z}) \right) \right]_k, \tag{12}$$

where $\mathbf{Z}$ denotes the nodes feature output by SGAE, $[\cdot]_k$ refers to a computation specific to head k of a multihead attention module. The $Encoder$ and $Decoder$ modules follow the definitions below.

$$Encoder(\mathbf{Z}) = SAB^l(\mathbf{Z}, \mathbf{Z}), \tag{13}$$

$$Decoder(\mathbf{H}) = FF \left( SAB^h \left( PMA(\mathbf{H}), PMA(\mathbf{H}) \right) \right), \tag{14}$$

$$PMA(\mathbf{H}) = SAB \left( \mathbf{s}, FF(\mathbf{H}) \right), \tag{15}$$

$$SAB(\mathbf{E}, \mathbf{Y}) = \mathbf{B} + FF(\mathbf{B}), \tag{16}$$

$$\mathbf{B} = \mathbf{E} + MultiHead(\mathbf{E}, \mathbf{Y}, \mathbf{Y}), \tag{17}$$

Here, $\mathbf{H}$ is the $Encoder$ output. The $Encoder$ consists of $l$ classical multi-head attention blocks ($SAB$) that do not include positional encoding. The $Decoder$ includes a multi-head attention pooling block ($PMA$) (where $\mathbf{s}$ is an initial output vector generated by a learnable seed vector), followed by further processing through $h$ self-attention modules and a linear projection block ($FF$). $\mathbf{E}$ and $\mathbf{Y}$ are the inputs to the $SAB$ module, typically in matrix form.

### 3.7 PRE-TRAINING AND FINE-TUNING

**Pre-training.** All parameters participate in training during the pre-training stage. This stage utilizes only drug response-related data from cell lines. The loss function for the pre-training stage is defined as follows:

$$\mathcal{L}_{pre} = MSE(P_1(\mathcal{Z}_{pre}), \mathcal{Y}_c) - \text{KL} \left[ q(\mathbf{Z}|\mathbf{X}, \mathbf{A}) \| p(\mathbf{Z}) \right], \tag{18}$$

where $\mathcal{Z}_{pre}$ denotes drug response features of cell lines, which is output by Adaptive Readout function ($AR$) in pre-training model. $\mathcal{Y}_c$ denotes the labels of drug response in cell lines. $MSE(\cdot)$ represents the mean square error loss function, $P_1(\cdot)$ is the predictor. KL $[\cdot\|\cdot]$ is the Kullback-Leibler divergence. Here, $p(\mathbf{Z})$ is a standard Gaussian prior $\mathcal{N}(0,1)$, as commonly used in variational autoencoders.

**Fine-tuning.** During the fine-tuning stage, we transfer parameters from the pre-trained model to the fine-tuned model. Subsequently, all parameters in the fine-tuned model are frozen except for the adaptive readout function and prediction module. This stage utilizes only drug response-related data from patients. The training loss function is as follows:

$$\mathcal{L}_{fine} = MSE(P_2([\mathcal{Z}_{fine} \| \hat{\mathcal{Z}}_{pre}]), \mathcal{Y}_p), \tag{19}$$

where $\mathcal{Z}_{fine}$ denotes drug response features of patients, which is output by the adaptive readout ($AR$) function in the fine-tuning model. $\hat{\mathcal{Z}}_{pre}$ is the response feature obtained by inputting the drug data and patient data used to compute $\mathcal{Z}_{fine}$ into a pre-trained model (with all parameters frozen). $[\cdot \| \cdot]$ denotes concatenation. $\mathcal{Y}_p$ denotes the labels of drug response in patients. $P_2(\cdot)$ is the predictor.

During the fine-tuning stage, we effectively balance domain-invariant knowledge and domain-specific knowledge through trainable AR modules and feature concatenation strategies. This enables drug response representations to adapt to patient-specific biological variations while retaining pre-trained knowledge from cell lines. This approach also mitigates the risk of overfitting in small-sample learning (patient drug responses). For further explanation of this strategy, please refer to Appendix A.17.

## 4 EXPERIMENTS

### 4.1 EXPERIMENTAL SETUPS

**Dataset.** We utilized the same cancer cell lines and patient genomic characteristics as WISER (Shubham et al., 2024) (including expression data for 1,426 genes), and the drug Smiles sequences were obtained from PubChem (Kim et al., 2019). We collected 966 cancer cell line samples with drug response label (used in pre-training stage) from the DepMap portal (Ghandi et al., 2019) and 555 patient samples with drug response label (used in fine-tuning stage) from the TCGA database. Drug responses in cell lines were determined based on z-score values calculated from the area under the dose-response curve (AUDRC). A z-score value less than 0 was considered a positive response and greater than 0 was considered a negative response. For patients, responses were assessed based on the time to cancer recurrence after chemotherapy, with responses exceeding the median classified as positive and those below the median as negative. For specific data preprocessing methods and related details, please refer to (He et al., 2022). In the pre-training stage, we selected 20 drugs that were present in both the DepMap and TCGA. In the fine-tuning stage, due to the limited amount of labeled patient genomic profiles, we only selected five drugs suitable for fine-tuning training (these five drugs all contained at least 20 cases of patient response data). More detial of data in Appendix A.5.

**Evaluation protocol.** This study primarily observes how the DRP model utilizes large-scale drug-cell line response data as a 'proxy' to predict drug responses in patient data (with a smaller data scale) through transfer learning. First, the DRP model is trained using drug-cell line response data (source domain). Then, the model is fine-tuned using a small amount of patient drug response data (target domain), and finally used to predict patient-drug response labels. During the fine-tuning phase, patient drug response data is divided into training and testing sets at a 7:3 ratio. We use two commonly used metrics to evaluate the model's classification performance: area under the curve (AUC) and area under the precision-recall curve (AUPR).

**Baselines.** We have compared this method with WISER (Shubham et al., 2024), GANDALF (Jayagopal et al., 2025b), CODE-AE (He et al., 2022), VAEN (Jia et al., 2021), DAE (Vincent et al., 2008), DruID (Jayagopal et al., 2025a), drug2tme (Zhai & Liu, 2024), TransDRP (Liu & Li, 2025), PANCDR(Kim et al., 2024) and PREDICT-AI (Jayagopal et al., 2024). Additionally, this study compares domain adaptation techniques such as Celligner (Warren et al., 2021), Velodrome (Sharifi-Noghabi et al., 2021), Deep CORAL (Sun & Saenko, 2016), and DSN (MMD and DANN variants) (Bousmalis et al., 2016). To compare with cell line-based drug response prediction models, we also included patient data in the experimental results of DeepTTA (Jiang et al., 2022) and GraphCDR (Liu et al., 2022). Detailed pipelines for each method are provided in Appendix A.6.

### 4.2 EXPERIMENTAL RESULTS

**Performance comparison.** As shown in Table 1, DeepSADR significantly outperforms the baseline model for three drugs (Temozolomide, Sorafenib and Cisplatin), with superior AUC and AUPR scores; it remains competitive for the remaining two drugs (Fluorouracil and Gemcitabine). Compared to more recent models, our approach of constructing the model from a subsequence perspective has proven effective, improving the model's predictive performance and enhancing its interpretability. In addition to comparing with newer baseline models, we also directly applied the cell line-directed model to patient data for testing (i.e., without transfer learning adjustments), with results shown in Appendix A.7. All cell line models performed poorly in predicting the efficacy of five drugs, with AUC and AUPR scores decreasing by approximately 0.2–0.3 compared to the fine-tuned model, demonstrating the effectiveness of our transfer learning strategy. In models that only consider the differences in data distribution between cell lines and patients, even the best-performing GANDALF lags behind our DeepSADR, indicating that our strategy of treating the overall biological processes of drugs in cell lines and patients as the source domain and target domain for transfer learning is more effective. The results of all our methods were obtained using multiple random seeds to obtain the mean/standard deviation, as shown in Appendix A.9.

**Ablation study.** To investigate the necessity of each module in the model architecture, we conducted several comparative experiments on DeepSADR and its variants:

Table 1: Performance (AUC and AUPR scores) comparison of all methods for 5 clinical drugs

| Methods | Fluorouracil | | Temozolomide | | Sorafenib | | Gemcitabine | | Cisplatin | |
|---|---|---|---|---|---|---|---|---|---|---|
| | AUC↑ | AUPR↑ | AUC↑ | AUPR↑ | AUC↑ | AUPR↑ | AUC↑ | AUPR↑ | AUC↑ | AUPR↑ |
| **DeepSADR** | **0.805/0.056** | **0.821/0.023** | **0.870/0.026** | **0.886/0.029** | **0.957/0.037** | **0.978/0.024** | **0.719/0.057** | **0.702/0.022** | **0.927/0.027** | **0.922/0.021** |
| GANDALF | 0.793/0.031 | 0.740/0.006 | 0.791/0.017 | 0.782/0.011 | 0.811/0.020 | 0.795/0.062 | 0.709/0.026 | 0.697/0.016 | 0.852/0.071 | 0.813/0.011 |
| WISER | 0.715/0.036 | 0.741/0.023 | 0.760/0.006 | 0.786/0.019 | 0.727/0.007 | 0.728/0.024 | 0.649/0.037 | 0.652/0.002 | 0.781/0.007 | 0.796/0.020 |
| CODE-AE | 0.782/0.021 | 0.722/0.016 | 0.742/0.017 | 0.732/0.021 | 0.631/0.020 | 0.705/0.062 | 0.594/0.016 | 0.651/0.006 | 0.652/0.071 | 0.743/0.011 |
| VAEN | 0.633/0.157 | 0.585/0.100 | 0.648/0.035 | 0.632/0.162 | 0.600/0.021 | 0.668/0.112 | 0.526/0.087 | 0.618/0.223 | 0.694/0.049 | 0.698/0.065 |
| DAE | 0.591/0.066 | 0.573/0.066 | 0.685/0.013 | 0.668/0.105 | 0.485/0.053 | 0.613/0.046 | 0.530/0.036 | 0.511/0.048 | 0.522/0.087 | 0.581/0.096 |
| DruID | 0.635/0.092 | 0.654/0.034 | 0.645/0.027 | 0.634/0.037 | 0.614/0.055 | 0.624/0.034 | 0.664/0.062 | 0.638/0.045 | 0.637/0.076 | 0.623/0.048 |
| drug2tme | 0.619/0.080 | 0.646/0.073 | 0.675/0.009 | 0.662/0.012 | 0.641/0.053 | 0.621/0.054 | 0.621/0.057 | 0.602/0.055 | 0.614/0.048 | 0.632/0.037 |
| CORAL | 0.578/0.015 | 0.651/0.135 | 0.675/0.020 | 0.654/0.020 | 0.491/0.023 | 0.616/0.048 | 0.597/0.030 | 0.544/0.037 | 0.617/0.072 | 0.617/0.124 |
| VELODROME | 0.598/0.054 | 0.403/0.002 | 0.701/0.028 | 0.668/0.003 | 0.505/0.029 | 0.749/0.005 | 0.547/0.030 | 0.434/0.022 | 0.583/0.029 | 0.442/0.012 |
| CELLIGNER | 0.536/0.060 | 0.531/0.024 | 0.454/0.070 | 0.454/0.070 | 0.454/0.070 | 0.575/0.029 | 0.520/0.053 | 0.497/0.042 | 0.550/0.033 | 0.575/0.029 |
| DSN-DANN | 0.635/0.065 | 0.596/0.101 | 0.683/0.015 | 0.690/0.040 | 0.533/0.050 | 0.628/0.069 | 0.555/0.070 | 0.582/0.044 | 0.585/0.103 | 0.608/0.133 |
| DSN-MMD | 0.678/0.074 | 0.674/0.103 | 0.712/0.031 | 0.759/0.051 | 0.515/0.036 | 0.582/0.090 | 0.465/0.041 | 0.491/0.069 | 0.650/0.023 | 0.605/0.067 |
| TransDRP | 0.791/0.013 | 0.794/0.113 | 0.721/0.021 | 0.715/0.035 | 0.731/0.033 | 0.766/0.082 | 0.635/0.014 | 0.598/0.042 | 0.665/0.027 | 0.648/0.036 |
| PREDICT-AI | 0.702/0.112 | 0.776/0.103 | 0.739/0.113 | 0.719/0.135 | 0.734/0.236 | 0.752/0.193 | 0.612/0.141 | 0.593/0.294 | 0.609/0.201 | 0.613/0.227 |
| PANCDR | 0.638/0.014 | 0.643/0.011 | 0.701/0.022 | 0.711/0.015 | 0.665/0.036 | 0.674/0.071 | 0.623/0.043 | 0.618/0.059 | 0.635/0.023 | 0.613/0.026 |
| DeepTTA | 0.569/0.050 | 0.599/0.042 | 0.646/0.022 | 0.624/0.038 | 0.444/0.035 | 0.501/0.035 | 0.467/0.036 | 0.498/0.049 | 0.459/0.070 | 0.496/0.070 |
| GraphCDR | 0.536/0.012 | 0.540/0.007 | 0.576/0.006 | 0.568/0.014 | 0.592/0.005 | 0.549/0.021 | 0.538/0.008 | 0.554/0.010 | 0.550/0.000 | 0.542/0.009 |

*Note:* Data related to clinical relapse is used for all evaluations. The results are reported as the mean/standard deviation of multiple random seeds. Best performer among all baselines is in **bold**.

- DeepSADR (w/o AR) that removes the adaptive readout module, replacing it with conventional readout functions (sum/max/mean).
- DeepSADR (w/o SN) that removes the subsequence interaction module and directly readout subsequence features through an adaptive readout function.
- DeepSADR (w/o TS) removes the threshold selection operation, it does not remove edges with lower weights (noise) in the subsequence interaction graph.
- DeepSADR (w/o ET) does not incorporate the pre-trained drug response features as additional input to the fine-tuning stage.

The ablation experiment results are shown in Table 2. When the readout function was replaced with a standard readout function (w/o AR), the AUC and AUPR scores decreased from 0.856 to 0.662 and from 0.862 to 0.675, respectively, indicating that the adaptive readout function we used has significant value for the DRP transfer learning task. The results of the variant (w/o ET) indicate that using the drug response features from the pre-training stage as additional input for the fine-tuning model does indeed facilitate transfer learning for DRP.

Table 2: Ablation results (average of 5 drugs).

| Methods | AUC↑ | AUPR↑ |
|---|---|---|
| DeepSADR | 0.856 | 0.862 |
| DeepSADR(w/o AR) | 0.662 | 0.675 |
| DeepSADR(w/o SN) | 0.698 | 0.710 |
| DeepSADR(w/o TS) | 0.775 | 0.749 |
| DeepSADR(w/o ET) | 0.781 | 0.787 |

The results of the variant (w/o SN) further confirm that converting the drug response process into a subsequence interaction graph can effectively improve the predictive performance of the DRP model and increase the model's interpretability. The results (w/o TS) indicate that removing some of the less significant connections edges (noise) from the subsequence interaction graph can improve the performance of the model. Overall, the DeepSADR model, which integrates the above modules, performs well, and the absence of any module will compromise its power (the ablation results for each drug, see Appendix A.8).

**Visualization analysis.** During the construction of the subsequence interaction graph, DeepSADR generates an interaction strength correlation graph to analyze the interaction strength between subsequences in the input drug and genomic profiles. Subsequences significantly contributing to drug response outcomes receive higher scores. To intuitively present the interaction relationships between these subsequences, we use a heat map (as shown in Figure 2) to display the interaction graph.

The Y-axis indicates which generalized functional pathway clusters (biological processes such as apoptosis, cycle regulation, etc.) are affected by temozolomide. The X-axis represents all substructure sequences (chemical fragments) of the temozolomide molecule. The intensity of colors in the heatmap indicates the weight of influence exerted by corresponding drug subsequences on biological functional pathway clusters. The sequence with the highest weight is highlighted in red. It

breaks through the limitations of other models that merely conclude "this drug is effective", delving into the deeper molecular mechanisms of "which structural component of this drug is effective and why", This not only thoroughly elucidates the mechanism of action of temozolomide but also provides a direct, visual chemical blueprint for designing next-generation, superior anticancer drugs.

To validate the reliability of the subsequence associations learned by our model, we conducted correlation validation on drug/gene subsequences with higher weights. Studies such as (Stupp et al., 2005; Zhang et al., 2012; Hegi et al., 2005) indicate that the "tetraene ring" fragment exhibits a high association score with the "cell stress and apoptosis pathway"—not merely a computational result, but a conclusion validated through extensive biological experiments. This demonstrates that our DeepSADR model accurately captures the strength of genuine sub-sequence interactions.

We also performed similar visualization analyses on other drugs, as well as other experiments such as parameter sensitivity analyses. For more details on other experiments and model parameters, please refer to the appendix A.10-A.20.

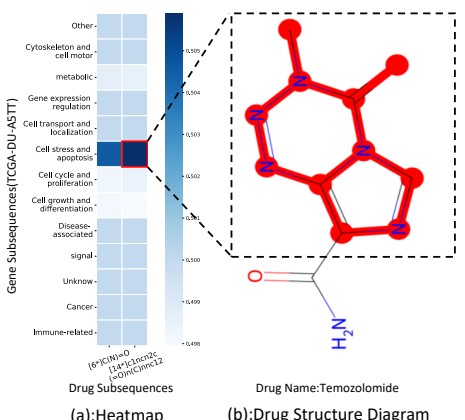

Figure 2: Visual analysis of patient-drug subsequence interactions. (a) shows a heat map. (b) shows the drug molecular structure.

## 5 CONCLUSION

In this study, we leverage biological knowledge within the framework of transfer learning to propose DeepSADR, a robust deep learning model for transferring drug responses from cell lines to clinical patients. On one hand, it converts drug responses into subsequence interaction graphs for feature extraction, enhancing predictive performance and model interpretability. On the other hand, it treats the entire drug response process as a domain, introducing an adaptive readout function to improve domain transfer accuracy. In transfer learning experiments from cell lines to the patients, DeepSADR outperformed other state-of-the-art methods in terms of predictive performance. However, DeepSADR has the following limitations: (1) Threshold tuning is required for each drug (potentially related to pharmacological diversity); (2) Performance depends to some extent on the quality of constructed sub-sequence interaction graphs. Future research should consider addressing these two aspects.

**REPRODUCIBILITY.** Our source code and the data used in our experiments are publicly available at https://github.com/ZYPssss/DeepSADR.

**ACKNOWLEDGMENTS.** This research was supported by the National Natural Science Foundation of China (Grant Nos. U23A20321 and 62272490); the Natural Science Foundation of Hunan Province of China (Grant No. 2025JJ20062); and A*STAR's Decentralised Gap Funding (I23D1AG081) and AIDD Catalyst Grant (H25A1N0003).

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

## 6 ETHICS STATEMENT

This work adheres to the ICLR Code of Ethics. In this study, no human subjects or animal experimentation was involved. All datasets used were sourced in compliance with relevant usage guidelines, ensuring no violation of privacy. We have taken care to avoid any biases or discriminatory outcomes in our research process. No personally identifiable information was used, and no experiments were conducted that could raise privacy or security concerns. We are committed to maintaining transparency and integrity throughout the research process.

## 7 REPRODUCIBILITY STATEMENT

We have made every effort to ensure that the results presented in this paper are reproducible. The experimental setup, including training steps, model configurations, and hardware details, is described in detail in the paper.

Additionally, all datasets used in this paper are publicly available resources, ensuring the consistency and reproducibility of the evaluation results.

We believe these measures will enable other researchers to reproduce our work and further advance the field.

## 8 LLM USAGE

Large Language Models (LLMs) were used to aid in the writing and polishing of the manuscript. Specifically, we used an LLM to assist in refining the language, improving readability, and ensuring clarity in various sections of the paper. The model helped with tasks such as sentence rephrasing, grammar checking, and enhancing the overall flow of the text.

It is important to note that the LLM was not involved in the ideation, research methodology, or experimental design. All research concepts, ideas, and analyses were developed and conducted by the authors. The contributions of the LLM were solely focused on improving the linguistic quality of the paper, with no involvement in the scientific content or data analysis.

The authors take full responsibility for the content of the manuscript, including any text generated or polished by the LLM. We have ensured that the LLM-generated text adheres to ethical guidelines and does not contribute to plagiarism or scientific misconduct.

## A APPENDIX

### A.1 PSEUDOCODE AND LIST OF NOTATIONS

This section shows the pseudocode of the DeepSADR in Algorithm1 and list of notations in Table 3.

---

**Algorithm 1** DeepSADR: Drug Response Prediction

---

**Require:** $D$: Drug set with SMILES sequences $\mathcal{S}(d_i)$
    $\mathcal{G}^c$: Genomic profiles of cell lines
    $\mathcal{G}^p$: Genomic profiles of patients
    $\mathcal{Y}_c$: Drug response labels for cell lines
    $\mathcal{Y}_p$: Drug response labels for patients
    $t$: Threshold for adjacency matrix (default=0.5)
1: **Stage 1: Pre-training on Cell Line Data**
2: **for** each drug $d_i \in D$ and cell line $g_j^c \in \mathcal{G}^c$ **do**
3:     **Step 1: Subsequence Decomposition**
4:     $[\mathcal{S}_{sub}^1, \ldots, \mathcal{S}_{sub}^n] \leftarrow \text{BRICS}(\mathcal{S}(d_i))$ {Drug SMILES decomposition}
5:     $[\mathcal{G}_{sub}^1, \ldots, \mathcal{G}_{sub}^m] \leftarrow \text{gseapy}(g_j^c)$ {Gene enrichment analysis}
6:     **Step 2: Feature Extraction**
7:     $Sub_i^d \leftarrow \{\text{GNP}(\mathcal{S}_{sub}^1), \ldots, \text{GNP}(\mathcal{S}_{sub}^n)\}$ {Drug subsequence features}
8:     $Sub_j^c \leftarrow \{f_1(\mathcal{G}_{sub}^1), \ldots, f_m(\mathcal{G}_{sub}^m)\}$ {Gene subsequence features}
9:     **Step 3: Subsequence Interaction Graph**
10:     $\mathcal{R}[i][j] \leftarrow \sigma(\text{GNP}(\mathcal{S}_{sub}^i) \cdot W \cdot f_j(\mathcal{G}_{sub}^j)^\top)$ {Interaction scores}
11:     $\hat{\mathcal{R}}[i][j] \leftarrow \begin{cases} \mathcal{R}[i][j] & \text{if } \mathcal{R}[i][j] \geq t \\ 00 & \text{otherwise} \end{cases}$ {Threshold filtering}
12:     Construct adjacency matrix $\mathbf{A}$ from $\hat{\mathcal{R}}$
13:     $\mathbf{Z} \leftarrow \text{SGAE}(\mathbf{X} = \{Sub_i^d, Sub_j^c\}, \mathbf{A})$ {Graph encoding}
14:     **Step 4: Adaptive Readout & Prediction**
15:     $\mathcal{Z} \leftarrow \frac{1}{K} \sum_{k=1}^K \text{Decoder}(\text{Encoder}(\mathbf{Z}))_k$ {Set Transformer readout}
16:     $\hat{\mathcal{Y}}_c \leftarrow P_1(\mathcal{Z})$ {Prediction}
17:     $\mathcal{L}_{pre} \leftarrow \text{MSE}(\hat{\mathcal{Y}}_c, \mathcal{Y}_c) - \text{KL}(q(\mathbf{Z}|\mathbf{X}, \mathbf{A})\|p(\mathbf{Z}))$ {Loss with VAE regularization}
18: **end for**
19: **Stage 2: Fine-tuning on Patient Data**
20: Freeze all layers except Adaptive Readout and $P_2$
21: **for** each drug $d_i \in D$ and patient $g_k^p \in \mathcal{G}^p$ **do**
22:     **Step 1: Forward through Pre-trained Model**
23:     Compute $\hat{\mathcal{Z}}_{pre}$ using frozen pre-trained model {Pre-trained features}
24:     **Step 2: Fine-tuned Forward Pass**
25:     $\mathcal{Z}_{fine} \leftarrow \text{Adaptive Readout}(\mathbf{Z})$ {Fine-tuned features}
26:     **Step 3: Feature Fusion & Prediction**
27:     $\hat{\mathcal{Y}}_p \leftarrow P_2([\mathcal{Z}_{fine}\|\hat{\mathcal{Z}}_{pre}])$ {Concatenate features}
28:     $\mathcal{L}_{fine} \leftarrow \text{MSE}(\hat{\mathcal{Y}}_p, \mathcal{Y}_p)$ {Fine-tuning loss}
29: **end for**
**Ensure:** Fine-tuned model for patient drug response prediction

---

### A.2 DISTINCTION BETWEEN THE TWO DOMAINS(CELL LINES AND PATIENTS)

Table 4 provides detailed information on the two study subjects (cell lines and patients) considered in this study. The cell line domain has rich labeled responses for multiple drugs, while patient data has only a small number of labeled samples. In our experiments, we selected 20 drugs that are used in both patients and cell lines. To assess the applicability of our method in patients, we considered five drugs with recorded responses in at least 20 patients.(The cell lines genomic profiles data can get in https://depmap.org/portal/ and the patients genomic profiles data can get in https://portal.gdc.cancer.gov/.)

Table 3: LIST OF NOTATIONS

| Symbol | Description |
|--------|-------------|
| $N$ | The number of drug |
| $N_c$ | The number of drug response data in cell lines |
| $N_p$ | The number of drug response data in patients |
| $\mathcal{G}^c$ | cell lines genomic profiles |
| $\mathcal{G}^p$ | patients genomic profiles |
| $d_i$ | The i-th drug |
| $g_j^c / g_j^p$ | The j-th genomic profiles of cell lines/patients |
| $\mathcal{S}_{sub}$ | The subsequence of drug Smiles |
| $\mathcal{G}_{sub}$ | The subsequence of genomic profiles |
| $Sub_i^d$ | The set of all subsequence features of drug i |
| $Sub_j^c / Sub_j^p$ | The set of all subsequence features of cell lines / patient j |
| $n$ | Indicates the number of subsequences of a drug decomposition |
| $m$ | Indicates the number of sub-sequences of a gene decomposition |
| $\mathcal{R} \in \mathbf{R}^{n \times m}$ | Subsequence interaction map |
| $\mathbf{Z}$ | The nodes feature output by SGAE |
| $\mathcal{Z}$ | The drug response feature output by adaptive readout functional |

Table 4: Details about the two domains in cancer drug response prediction.

| Domains | Used stage | Labeled data | Drug response label | Number of drug responses | Number of drugs selected in our experiments |
|---------|-----------|-------------|---------------------|-------------------------|-------------------------------------------|
| Cell lines | Pre-training | 225,781 | Evaluated using Z-score computed on AUDRC scores. (1) Z-score less than 0 considered as positive respondents. (2) Z-score greater than 0 considered as negative respondents. | 11,538 | 20 |
| Patients | Fine-tuning | 576 | Cancer relapse time post-chemotherapy (1) Values greater than the median considered positive respondents. (2) Values less than the median considered negative respondents. | 338 | 5 |

## A.3 THE DETAILS OF GNP

To obtain better drug sub-sequence features, we used Wu et al.'s method to improve its performance in graph-level tasks. The scheme integrates six popular techniques: edge feature fusion, normalization, dropout, residual connections, feedforward networks (FFN), and position encoding.

### A.3.1 EDGE FEATURE INTEGRATION

Edge features were initially incorporated into GNN frameworks (Gilmer et al., 2017) by integrating them into the message-passing process. Following this practice, we encode edge features to enrich node representations. For GCN (Kipf & Welling, 2016b):

$$\boldsymbol{h}_v^l = \sigma\Big( \sum_{u \in \mathcal{N}(v) \cup \{v\}} \frac{1}{\sqrt{\hat{d}_u \hat{d}_v}} \boldsymbol{h}_u^{l-1} \boldsymbol{W}^l + \boldsymbol{e}_{uv} \boldsymbol{W}_e^l \Big), \tag{20}$$

where $\boldsymbol{W}_e^l$ is the trainable weight matrix for edges.

### A.3.2 NORMALIZATION

Normalization techniques stabilize GNN training by mitigating covariate shift. We use Batch Normalization (Ioffe & Szegedy, 2015):

$$\boldsymbol{h}_v^l = \sigma(BN(\sum_{u \in \mathcal{N}(v) \cup \{v\}} \frac{1}{\sqrt{\hat{d}_u \hat{d}_v}} \boldsymbol{h}_u^{l-1} \boldsymbol{W}^l + \boldsymbol{e}_{uv} \boldsymbol{W}_e^l)). \tag{21}$$

### A.3.3 DROPOUT

Dropout (Srivastava et al., 2014) addresses overfitting by reducing co-adaptation among neurons. Applied after activation:

$$\boldsymbol{h}_v^l = \text{Dropout}(\sigma (BN(\sum_{u \in \mathcal{N}(v) \cup \{v\}} \frac{1}{\sqrt{\hat{d}_u \hat{d}_v}} \boldsymbol{h}_u^{l-1} \boldsymbol{W}^l + \boldsymbol{e}_{uv} \boldsymbol{W}_e^l))). \tag{22}$$

### A.3.4 RESIDUAL CONNECTION

Residual connections (He et al., 2016) alleviate vanishing gradients. Integrated as:

$$\boldsymbol{h}_v^l = \text{Dropout}(\sigma(BN(\sum_{u \in \mathcal{N}(v) \cup \{v\}} \frac{1}{\sqrt{\hat{d}_u \hat{d}_v}} \boldsymbol{h}_u^{l-1} \boldsymbol{W}^l \tag{23}$$

$$+ \boldsymbol{e}_{uv} \boldsymbol{W}_e^l))) + \boldsymbol{h}_v^{l-1}. \tag{24}$$

### A.3.5 FEED-FORWARD NETWORK

Inspired by Transformers (Vaswani et al., 2017), we append an FFN:

$$FFN(\boldsymbol{h}) = BN(\sigma(\boldsymbol{h} \boldsymbol{W}_{\text{FFN}_1^l}) \boldsymbol{W}_{\text{FFN}_2^l} + \boldsymbol{h}), \tag{25}$$

The node embeddings become:

$$\boldsymbol{h}_v^l = FFN(\text{Dropout}(\sigma(BN(\sum_{u \in \mathcal{N}(v) \cup \{v\}} \frac{1}{\sqrt{\hat{d}_u \hat{d}_v}} \boldsymbol{h}_u^{l-1} \boldsymbol{W}^l \tag{26}$$

$$+ \boldsymbol{e}_{uv} \boldsymbol{W}_e^l))) + \boldsymbol{h}_v^{l-1}). \tag{27}$$

### A.3.6 POSITIONAL ENCODING

We use Random Walk Structural Encoding (RWSE) (Li et al., 2020):

$$\boldsymbol{x}_v = [\boldsymbol{x}_v \parallel \boldsymbol{x}_v^{RWSE}] \boldsymbol{W}_{\text{PE}}, \tag{28}$$

where $[\cdot \parallel \cdot]$ denotes concatenation and $\boldsymbol{W}_{\text{PE}}$ is trainable.

## A.4 SUPERVISED GRAPH AUTOENCODER

We use supervised graph autoencoder to further extract features from the sub-sequence interaction graph. A graph autoencoder consists of a probabilistic encoder and decoder, with several important differences compared to standard architectures operating on vector-valued inputs. The encoder component is obtained by stacking graph convolutional layers to learn the parameter matrices $\boldsymbol{\mu}$ and $\boldsymbol{\sigma}$ that specify the Gaussian distribution of a latent space encoding.

$$q(\mathbf{Z}|\mathbf{X}, \mathbf{A}) = \prod_{i=1}^{n+m} q(z_i|\mathbf{X}, \mathbf{A}), \tag{29}$$

$$q(z_i|\mathbf{X}, \mathbf{A}) = \mathcal{N}\left(z_i | \boldsymbol{\mu}_i, \text{diag}(\boldsymbol{\sigma}_i^2)\right), \tag{30}$$

Here, $\boldsymbol{\mu} = GCN_{\boldsymbol{\mu}}(\mathbf{X}, \mathbf{A})$ is the matrix of mean vectors $\boldsymbol{\mu}_i$, similarly $log\boldsymbol{\sigma} = GCN_{\boldsymbol{\sigma}}(\mathbf{X}, \mathbf{A})$. The two-layer GCN is defined as $GCN(\mathbf{X}, \mathbf{A}) = \tilde{\mathbf{A}} ReLU(\tilde{\mathbf{A}} \mathbf{X} \mathbf{W}_0) \mathbf{W}_1$, with weight matrices $\mathbf{W}_i$. $GCN_{\boldsymbol{\mu}}(\mathbf{X}, \mathbf{A})$ and $GCN_{\boldsymbol{\sigma}}(\mathbf{X}, \mathbf{A})$ share first-layer parameters $W_0$. $ReLU(\cdot) = max(0, \cdot)$ and $\tilde{\mathbf{A}} = \mathbf{D}^{-1/2} \mathbf{A} \mathbf{D}^{1/2}$ is the symmetrically normalized adjacency matrix. $\mathcal{N}$ denotes the Gaussian distribution.

Table 5: Annotated samples of the 5 drugs

| Drug name | samples | Pos | Neg | train_num | test_num |
|---|---|---|---|---|---|
| Fluorouracil | 88 | 47 | 41 | 52 | 36 |
| Cisplatin | 40 | 20 | 20 | 24 | 16 |
| Sorafenib | 26 | 13 | 13 | 15 | 11 |
| Gemcitabine | 138 | 60 | 78 | 82 | 56 |
| Temozolomide | 46 | 23 | 23 | 27 | 19 |

Table 6: Performance (AUC and AUPR scores) comparison of all methods for 5 clinical drugs

| Methods | Fluorouracil | | Temozolomide | | Sorafenib | | Gemcitabine | | Cisplatin | |
|---|---|---|---|---|---|---|---|---|---|---|
| | AUC↑ | AUPR↑ | AUC↑ | AUPR↑ | AUC↑ | AUPR↑ | AUC↑ | AUPR↑ | AUC↑ | AUPR↑ |
| **DeepSADR** | **0.805/0.056** | **0.821/0.023** | **0.870/0.026** | **0.886/0.029** | **0.957/0.037** | **0.978/0.024** | **0.719/0.057** | **0.702/0.022** | **0.927/0.027** | **0.922/0.021** |
| GANDALF | 0.793/0.031 | 0.740/0.006 | 0.791/0.017 | 0.782/0.011 | 0.811/0.020 | 0.795/0.062 | 0.709/0.026 | 0.697/0.016 | 0.852/0.071 | 0.813/0.011 |
| WISER | 0.715/0.036 | 0.741/0.023 | 0.760/0.006 | 0.786/0.019 | 0.727/0.007 | 0.728/0.024 | 0.649/0.037 | 0.652/0.002 | 0.781/0.007 | 0.796/0.020 |
| CODE-AE | 0.782/0.021 | 0.722/0.016 | 0.742/0.017 | 0.732/0.021 | 0.631/0.020 | 0.705/0.062 | 0.594/0.016 | 0.651/0.006 | 0.652/0.071 | 0.743/0.011 |
| VAEN | 0.633/0.157 | 0.585/0.100 | 0.648/0.035 | 0.632/0.162 | 0.600/0.021 | 0.668/0.112 | 0.526/0.087 | 0.618/0.223 | 0.694/0.049 | 0.698/0.065 |
| DAE | 0.591/0.066 | 0.573/0.066 | 0.685/0.013 | 0.668/0.105 | 0.485/0.053 | 0.613/0.046 | 0.530/0.036 | 0.511/0.048 | 0.522/0.087 | 0.581/0.096 |
| DruID | 0.635/0.092 | 0.654/0.034 | 0.645/0.027 | 0.634/0.037 | 0.614/0.055 | 0.624/0.034 | 0.664/0.062 | 0.638/0.045 | 0.637/0.076 | 0.623/0.048 |
| drug2tme | 0.619/0.080 | 0.646/0.073 | 0.675/0.009 | 0.662/0.012 | 0.641/0.053 | 0.621/0.054 | 0.621/0.057 | 0.602/0.055 | 0.614/0.048 | 0.632/0.037 |
| CORAL | 0.578/0.015 | 0.651/0.135 | 0.675/0.020 | 0.654/0.020 | 0.491/0.023 | 0.616/0.048 | 0.597/0.030 | 0.544/0.037 | 0.617/0.072 | 0.617/0.124 |
| VELODROME | 0.598/0.054 | 0.403/0.002 | 0.701/0.028 | 0.668/0.003 | 0.505/0.029 | 0.749/0.005 | 0.547/0.030 | 0.434/0.022 | 0.583/0.029 | 0.442/0.012 |
| CELLIGNER | 0.536/0.060 | 0.531/0.024 | 0.454/0.070 | 0.454/0.070 | 0.454/0.070 | 0.575/0.029 | 0.520/0.053 | 0.497/0.042 | 0.550/0.033 | 0.575/0.029 |
| DSN-DANN | 0.635/0.065 | 0.596/0.101 | 0.683/0.015 | 0.690/0.040 | 0.533/0.050 | 0.628/0.069 | 0.555/0.070 | 0.582/0.044 | 0.585/0.103 | 0.608/0.133 |
| DSN-MMD | 0.678/0.074 | 0.674/0.103 | 0.712/0.031 | 0.759/0.051 | 0.515/0.036 | 0.582/0.090 | 0.465/0.041 | 0.491/0.069 | 0.650/0.023 | 0.605/0.067 |
| TransDRP | 0.791/0.013 | 0.794/0.113 | 0.721/0.021 | 0.715/0.035 | 0.731/0.033 | 0.766/0.082 | 0.635/0.014 | 0.598/0.042 | 0.665/0.027 | 0.648/0.036 |
| PREDICT-AI | 0.702/0.112 | 0.776/0.103 | 0.739/0.113 | 0.719/0.135 | 0.734/0.236 | 0.752/0.193 | 0.612/0.141 | 0.593/0.294 | 0.609/0.201 | 0.613/0.227 |
| PANCDR | 0.638/0.014 | 0.643/0.011 | 0.701/0.022 | 0.711/0.015 | 0.665/0.036 | 0.674/0.071 | 0.623/0.043 | 0.618/0.059 | 0.635/0.023 | 0.613/0.026 |
| DeepTTA | 0.569/0.050 | 0.599/0.042 | 0.646/0.022 | 0.624/0.038 | 0.444/0.035 | 0.501/0.035 | 0.467/0.034 | 0.498/0.049 | 0.459/0.070 | 0.496/0.070 |
| GraphCDR | 0.536/0.012 | 0.540/0.007 | 0.576/0.006 | 0.568/0.014 | 0.592/0.005 | 0.549/0.021 | 0.538/0.008 | 0.554/0.010 | 0.550/0.000 | 0.542/0.009 |

*Note:* Data related to clinical relapse is used for all evaluations. The results are reported as the mean/standard deviation of multiple random seeds. Best performer among all baselines is in **bold**.

## A.5 DETAILS OF DATA USED IN THE EXPERIMENT

Table 5 shows the data details of the five drugs used in the fine-tuning phase. It is worth noting that both the DepMap and TCGA datasets we utilized underwent standardized preprocessing pipelines, including log transformation and quantile normalization, to mitigate batch effects. Furthermore, we adhered to the same gene expression normalization as WISER (Shubham et al., 2024) to ensure consistency.

## A.6 THE INTRODUCTION OF BASELINES

The following is an introduction to the baselines we selected for our comparative experiments:

- WISER (Shubham et al., 2024). A weakly supervised and supervised representation learning fusion framework aimed at solving the problem of scarce labeled data in cancer drug response prediction. By integrating unlabeled omics data (such as gene expression and mutations) to generate pseudo labels, and combining them with labeled data to jointly train the model, the robustness of prediction in small sample scenarios is improved.

- GANDALF (Jayagopal et al., 2025b). This is a generative attention framework for predicting cancer drug responses. It employs a diffusion model and cross-domain attention mechanism to directly transform cell line data into augmented samples that retain key information while exhibiting distributions closer to real patients. By generating pseudo labels through multi-task learning, it effectively enhances personalized prediction performance in data-scarce scenarios.

- CODE-AE (He et al., 2022). Context-decoupled autoencoder. Separate biological background information (such as cell type) from drug response-specific features in omics data through adversarial training to eliminate confounding factors. The decoder reconstructs samples based on decoupled features, while the predictor focuses on drug response signals, significantly improving clinical translation capabilities.

- VAEN (Jia et al., 2021). Variational autoencoder network. The VAE architecture is used to learn the low-dimensional manifold structure of gene expression data. The generator reconstructs the input features, and the predictor infers drug sensitivity based on latent variables.

- DAE (Vincent et al., 2008). Classic denoising autoencoder. By adding noise to the input data and training the network to reconstruct the original signal, it learns robust feature representations. As an early deep learning method, it provides a basic feature extraction module for subsequent drug prediction models, and is widely used in pre-training with unlabeled data.

- DruID (Jayagopal et al., 2025a). Multi-task domain adaptation model. For cancer recurrence prediction, a shared encoder is designed to learn general gene features across tumor types, while branch decoders are adapted to different chemotherapy drugs. MMD loss is used to align the source domain (cell lines) and target domain (patients) distributions to mitigate domain shift issues.

- drug2tme (Zhai & Liu, 2024). Tumor microenvironment decoupling framework. Separate tumor microenvironment (TME)-related features from cancer cell intrinsic features using graph neural networks, and quantify the impact of TME on drug efficacy. Provide interpretable analysis of drug response mechanisms to guide combination therapy design.

- Celligner (Warren et al., 2021). Cell Line-Patient Transcriptome Alignment Tool. Calculates gene expression similarity between cell lines and patient tumors based on optimal transport theory, constructing a cross-domain mapping matrix to correct biological discrepancies between preclinical models and real patients.

- Velodrome (Sharifi-Noghabi et al., 2021). Distribution-based generalization framework. Combining labeled and unlabeled data, distribution-robust features are learned through domain-invariant regularization (such as MMD and CORAL) and adversarial training. Specializing in drug response prediction for unknown cancer subtypes, it has been verified to outperform traditional methods in TCGA pan-cancer data.

- Deep CORAL (Sun & Saenko, 2016). Classic domain adaptation method. Align the second-order statistics (covariance matrix) of the source domain and target domain to minimize the difference between domains.

- DSN (MMD and DANN variants) (Bousmalis et al., 2016). Domain separation network. Contains private encoders and shared encoders: extracts domain-invariant features and difference losses, and separates public and private features through MMD or adversarial training. The effectiveness of feature decoupling has been verified in cross-domain drug sensitivity prediction.

- TransDRP (Liu & Li, 2025). Proposes a knowledge-guided domain-adaptive model that simultaneously predicts multiple drug responses and captures their correlations via multi-label graph neural networks. It employs a global-local adversarial strategy to achieve feature alignment and separation across cancer types, thereby enabling the transfer of drug response predictions from cell lines to patients.

- PANCDR (Kim et al., 2024). Constructed a two-stage adversarial network that alternately trains the discriminator and predictor models (rather than gradient reversal) to reduce the distribution discrepancy between preclinical and clinical data. It employs a Gaussian encoder to extract gene expression features, enabling patient-specific prediction of anticancer drug responses.

- PREDICT-AI (Jayagopal et al., 2024). Designed a Transformer-based sequence modeling approach that directly processes variable-length mutation sequences from clinical diagnostic panels through a novel annotation strategy. It integrates patient survival information as an auxiliary supervisory signal to enhance the accuracy of drug response prediction.

- DeepTTA (Jiang et al., 2022). Substructure encoding of drug molecule SMILES sequences and cell line gene expression as feature vectors are used to model drug-target-genome interactions through a multi-head attention mechanism. Transformer is introduced for the first time into IC50 prediction, achieving SOTA on the GDSC/CCLE dataset.

- GraphCDR (Liu et al., 2022). Graph contrastive learning model. Graph neural networks are used to aggregate neighborhood information, and a contrastive loss function is designed

to maximize the similarity of positive samples (similar drug sensitivity) and minimize the similarity of negative samples. This model performs exceptionally well in cold-start drug prediction.

Table 7: Results of five drugs on the unfine-tuned DeepSADR model.

| Drug | fine-tuning | | no fine-tuning | |
|---|---|---|---|---|
| | AUC↑ | AUPR↑ | AUC↑ | AUPR↑ |
| Fluorouracil | 0.806 | 0.821 | 0.534 | 0.557 |
| Cisplatin | 0.927 | 0.922 | 0.585 | 0.675 |
| Sorafenib | 0.957 | 0.978 | 0.609 | 0.621 |
| Gemcitabine | 0.719 | 0.702 | 0.496 | 0.501 |
| Temozolomide | 0.870 | 0.886 | 0.617 | 0.667 |

## A.7 EXPERIMENTAL RESULTS OF DEEPSADR WITHOUT FINE-TUNING

To verify that our model achieved effective transfer learning through fine-tuning strategies, we predicted the responses of five drugs in patients before fine-tuning the model. The results are shown in Table 7.

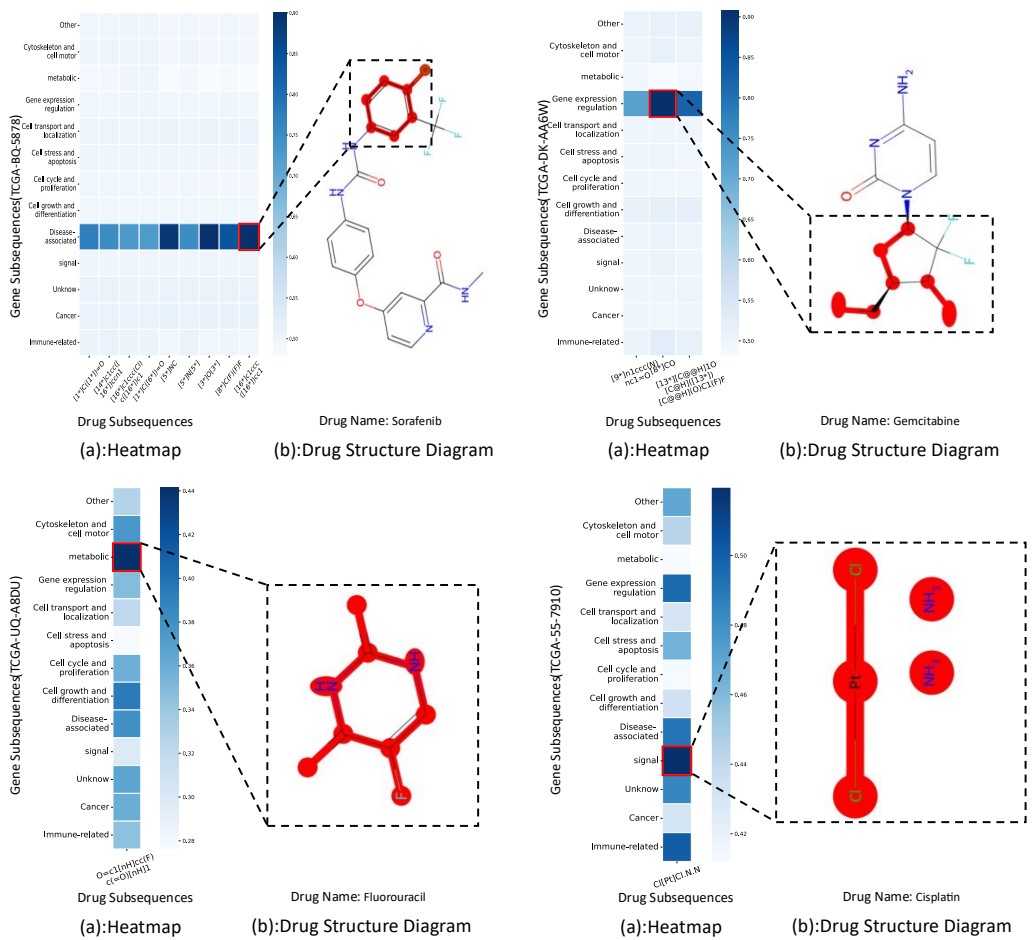

Figure 3: Visualization of subsequence interactions.

Table 8: Results of ablation experiments conducted on five different drugs

| Methods | Fluorouracil | | Temozolomide | | Sorafenib | | Gemcitabine | | Cisplatin | |
|---|---|---|---|---|---|---|---|---|---|---|
| | AUC↑ | AUPR↑ | AUC↑ | AUPR↑ | AUC↑ | AUPR↑ | AUC↑ | AUPR↑ | AUC↑ | AUPR↑ |
| **DeepSADR** | **0.805/0.056** | **0.821/0.023** | **0.870/0.026** | **0.886/0.029** | **0.957/0.037** | **0.978/0.024** | **0.719/0.057** | **0.702/0.022** | **0.927/0.027** | **0.922/0.021** |
| DeepSADR(w/o AR) | 0.715/0.036 | 0.741/0.023 | 0.660/0.006 | 0.686/0.019 | 0.621/0.007 | 0.628/0.024 | 0.591/0.037 | 0.582/0.002 | 0.721/0.007 | 0.736/0.020 |
| DeepSADR(w/o SN) | 0.741/0.031 | 0.762/0.006 | 0.691/0.017 | 0.722/0.011 | 0.661/0.020 | 0.695/0.062 | 0.649/0.026 | 0.657/0.016 | 0.732/0.071 | 0.713/0.011 |
| DeepSADR(w/o TS) | 0.774/0.021 | 0.734/0.016 | 0.804/0.017 | 0.782/0.021 | 0.831/0.020 | 0.805/0.062 | 0.649/0.016 | 0.651/0.006 | 0.815/0.071 | 0.773/0.011 |
| DeepSADR(w/o ET) | 0.753/0.157 | 0.775/0.101 | 0.814/0.035 | 0.802/0.162 | 0.849/0.021 | 0.857/0.112 | 0.653/0.087 | 0.664/0.223 | 0.834/0.049 | 0.837/0.065 |
| DeepSADR(w/o GNP) | 0.784/0.039 | 0.809/0.003 | 0.858/0.017 | 0.861/0.087 | 0.907/0.016 | 0.915/0.027 | 0.703/0.071 | 0.691/0.033 | 0.889/0.057 | 0.887/0.054 |

*Note:* Data related to clinical relapse is used for all evaluations. The results are reported as the mean/standard deviation of multiple random seeds. Best performer among all baselines is in **bold**.

## A.8 MORE DETAILS OF ABLATION

To maintain consistency with the comparative experiment, we presented the ablation experiment results for the five drugs, as shown in Table 8. To validate the positive guidance of our model by introducing the GNP module for extracting drug sub-sequence features, we conducted an ablation experiment on the GNP module. Specifically, we replaced the GNP module with the simplest GNN to extract drug sub-sequence features. The experimental results are shown in the DeepSADR (w/o GNP) section of Table 8.

We found that using either the AR module or the SN module alone did not yield significant performance improvements. This observation was precisely the impetus for designing our model framework. Our analysis of this phenomenon is as follows:

DeepSADR (w/o SN) performs poorly, indicating that the structured information provided by the sub-sequence interaction graph is crucial. Without this information, the model degrades into a basic feature concatenation model. Many previous drug response models have made predictions solely by concatenating drug and gene transcription features. We found that transfer learning based on such features yields suboptimal results.

DeepSADR (w/o AR) underperforms, indicating that intelligent feature aggregation in interaction graphs is key to transfer learning. Simple aggregation operations (e.g., summation) lose substantial critical information. We reached this conclusion through molecular property prediction experiments.

The two modules synergize: the SN module generates high-quality, interpretable input representations, while the AR module extracts domain-relevant information for cross-domain tasks. Together, they address the dual core challenges of domain transfer and interpretability. We supplement this analysis in our paper.

## A.9 THE MORE DETAIL OF COMPARATIVE EXPERIMENT

In the comparative experiments in the main text, we only wrote the average values and did not write the standard deviation. This section supplements the standard deviation in the comparative experiments. See Table 6 for details.

## A.10 MORE DETAIL OF VISUALIZATION ANALYSIS.

Since we trained five drugs in fine-tuning, we visualized the sub-sequence interaction graphs of the five drugs. The results are shown in Figure 2.

In addition, we also visualized the response characteristics of the trained drugs in cell lines and patients, and the results are shown in Figure 4. For part (a) in Figure 4, we visualize using molecular fingerprint features and patient genetic data. For part (b), we visualize the concatenated features derived from the pre-trained feature $\hat{\mathcal{Z}}_{pre}$ and fine-tuned feature $\mathcal{Z}_{fine}$ of patient drug responses.

The results show that the feature distribution of the trained model exhibits good consistency, indicating that our model has achieved a certain degree of generalization of drug response distribution in cell lines and patients.

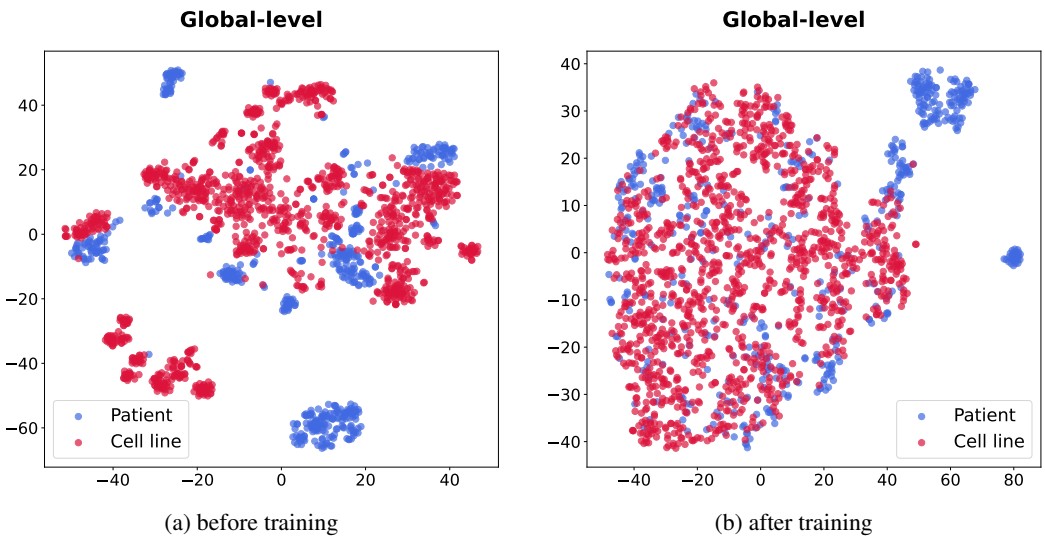

Figure 4: t-SNE visualization of the drug response feature in cell line data and patient data.

## A.11 PARAMETER SENSITIVITY ANALYSIS.

Since ablation studies have shown the importance of modules such as threshold selection and readout functions, we conducted sensitivity analyses on some of the more important parameters in these modules. Figure 5 shows the sensitivity experiments for four parameters: threshold parameter, convolution layer parameter, number of attention heads, and Dropout. From the results in the Figure 5, we find that the model's performance is quite sensitive to the threshold selection parameter. We analyzed that threshold selection is a critical part of constructing the sub-sequence interaction graph. Initially, we trained the model using the same threshold for each drug, but we found significant fluctuations in model performance across different drugs. We analyzed that this might be because different drugs have varying degrees of sensitivity to the threshold. A threshold that is too small may introduce excessive noise interference, while a threshold that is too large may remove important sub-sequence interaction information. This is an area we will further investigate in the future.

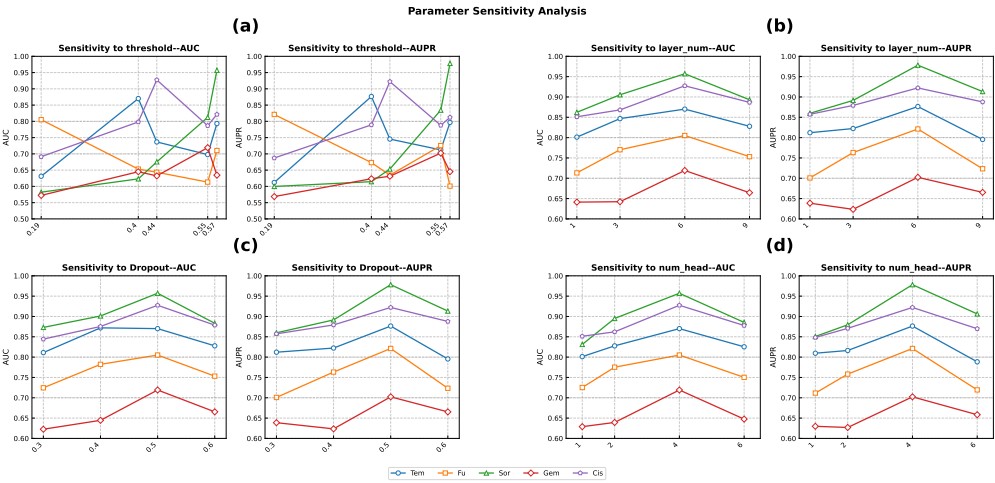

Figure 5: Parameter sensitivity analysis. (a) threshold parameter sensitivity analysis, (b) sensitivity analysis of the number of convolutional layers in SGAE, (c) sensitivity analysis of the number of attention heads in the adaptive readout function, and (d) sensitivity analysis of the dropout rate in the adaptive readout function.

Table 9: The detail params of five drugs in DeepSADR model.

| Drug name | threshold | convolution layer | attention heads | dropout | train_epoch | train_lr | fine_epoch | fine_lr |
|---|---|---|---|---|---|---|---|---|
| Fluorouracil | 0.19 | 6 | 4 | 0.5 | 200 | 1e-05 | 100 | 1e-04 |
| Cisplatin | 0.44 | 6 | 4 | 0.5 | 200 | 1e-05 | 100 | 1e-04 |
| Sorafenib | 0.57 | 6 | 4 | 0.5 | 200 | 1e-05 | 100 | 1e-04 |
| Gemcitabine | 0.55 | 6 | 4 | 0.5 | 200 | 1e-05 | 100 | 1e-04 |
| Temozolomide | 0.4 | 6 | 4 | 0.5 | 200 | 1e-05 | 100 | 1e-04 |

Table 10: Range of hyper-parameters used for optimizing model performance.

| Hyper-parameter | Range | Distribution |
|---|---|---|
| threshold | [0.1, 0.9] | Float (step size=0.01) |
| convolution layer | [1, 3, 6, 9] | Categorical |
| dropout | [0.3, 0.4, 0.5, 0.6] | Categorical |
| attention heads | [1, 2, 4, 6] | Categorical |
| fine_lr | $[1e^{-4}, 1e^{-1}]$ | Log uniform |
| train_lr | $[1e^{-4}, 1e^{-1}]$ | Log uniform |

Table 9 lists the parameter details for each drug to achieve the best predicted results. Tabel 10 shows the range of hyper-parameters used for optimizing DeepSADR model performance.

## A.12 PERFORMANCE ON OTHER DRUGS

In addition to testing the five drugs mentioned in the main text (for which relatively more known patient response data exists and positive and negative samples are balanced), we also conducted corresponding experiments on several other drugs with fewer response data. The results are shown in Table 11.

Table 11: Performance (AUC and AUPR scores) comparison of all methods for other clinical drugs

| Methods | Sunitinib | | Doxorubicin | | Erlotinib | |
|---|---|---|---|---|---|---|
| | AUC↑ | AUPR↑ | AUC↑ | AUPR↑ | AUC↑ | AUPR↑ |
| **DeepSADR** | **0.705/0.026** | **0.691/0.013** | **0.703/0.016** | **0.686/0.009** | **0.727/0.027** | **0.718/0.014** |
| GANDALF | 0.623/0.013 | 0.625/0.016 | 0.621/0.017 | 0.622/0.011 | 0.611/0.011 | 0.603/0.022 |
| WISER | 0.615/0.013 | 0.633/0.021 | 0.611/0.006 | 0.637/0.019 | 0.621/0.009 | 0.628/0.014 |
| CODE-AE | 0.582/0.011 | 0.552/0.012 | 0.562/0.027 | 0.532/0.031 | 0.591/0.021 | 0.605/0.032 |
| VAEN | 0.563/0.017 | 0.585/0.100 | 0.598/0.025 | 0.562/0.012 | 0.592/0.021 | 0.568/0.022 |
| DAE | 0.571/0.026 | 0.573/0.016 | 0.558/0.013 | 0.568/0.015 | 0.585/0.003 | 0.593/0.026 |
| DruID | 0.565/0.009 | 0.554/0.024 | 0.564/0.007 | 0.534/0.037 | 0.574/0.015 | 0.554/0.014 |
| drug2tme | 0.529/0.008 | 0.546/0.003 | 0.575/0.009 | 0.562/0.012 | 0.541/0.005 | 0.521/0.004 |
| CORAL | 0.518/0.015 | 0.551/0.035 | 0.517/0.002 | 0.565/0.007 | 0.491/0.023 | 0.616/0.048 |
| VELODROME | 0.518/0.004 | 0.493/0.002 | 0.571/0.018 | 0.568/0.003 | 0.515/0.029 | 0.549/0.005 |
| CELLIGNER | 0.536/0.060 | 0.531/0.024 | 0.454/0.070 | 0.454/0.070 | 0.454/0.070 | 0.575/0.029 |
| DSN-DANN | 0.535/0.005 | 0.526/0.021 | 0.501/0.005 | 0.511/0.004 | 0.503/0.050 | 0.518/0.009 |
| DSN-MMD | 0.508/0.004 | 0.521/0.003 | 0.492/0.003 | 0.518/0.005 | 0.485/0.006 | 0.509/0.009 |
| TransDRP | 0.581/0.031 | 0.589/0.013 | 0.612/0.012 | 0.613/0.014 | 0.573/0.023 | 0.576/0.023 |
| PREDICT-AI | 0.557/0.218 | 0.573/0.132 | 0.523/0.233 | 0.519/0.251 | 0.549/0.236 | 0.558/0.154 |
| PANCDR | 0.611/0.004 | 0.622/0.018 | 0.681/0.012 | 0.658/0.011 | 0.569/0.026 | 0.598/0.016 |
| DeepTTA | 0.519/0.005 | 0.529/0.012 | 0.546/0.012 | 0.524/0.003 | 0.484/0.015 | 0.501/0.005 |
| GraphCDR | 0.516/0.012 | 0.512/0.007 | 0.526/0.006 | 0.508/0.014 | 0.512/0.002 | 0.509/0.002 |

*Note:* Data related to clinical relapse is used for all evaluations. The results are reported as the mean/standard deviation of multiple random seeds. Best performer among all baselines is in **bold**.

Based on the experimental results, our model exhibited a certain decline in performance, primarily due to insufficient sample sizes for some drugs, which prevented fine-tuning (hence we directly input the data into the pre-trained model to generate corresponding outputs). Although our model's performance decreased, it still demonstrated certain advantages compared to all baseline models.

### A.13 Performance across different cancer types

To further validate our model's performance and demonstrate its ability to achieve effective transfer learning across the entire drug response spectrum, we conducted cross-cancer experiments. Patients with the same cancer type were grouped together, then separately fine-tuned and tested. The corresponding results are shown in Table 12.

Table 12: Comparison of performance across various cancer types.

| Cancer Type | AUC↑ | AUPR↑ |
|---|---|---|
| TCGA-CN | 0.874/0.036 | 0.862/0.032 |
| TCGA-2J | 0.857/0.042 | 0.864/0.036 |
| TCGA-IB | 0.762/0.037 | 0.798/0.037 |
| TCGA-VQ | 0.759/0.038 | 0.783/0.023 |
| TCGA-DU | 0.849/0.053 | 0.826/0.051 |

### A.14 Treating drug reactions as a distinct field

We treat drug responses as a distinct field because there are well-documented biological differences in how drugs act in preclinical models (such as cell lines) versus clinical patients. The notorious "translation gap" stems precisely from these disparities, including factors like the tumor microenvironment, immune regulation, and drug pharmacokinetics. Existing approaches primarily focus on revealing static, invariant features within the genome, yet fail to adequately model the dynamic, mechanistic variations occurring during the response process. Therefore, our core philosophy is this: domain differences manifest not only as variations in gene expression data distributions but fundamentally reflect inherent disparities in drug response mechanisms. By modeling the drug response process as a specific domain and leveraging adaptive readout technology to learn domain-invariant representations, we can more directly address this fundamental challenge.

To aid understanding, we can draw an analogy: Consider the process of diagnosing a car malfunction. Cell lines are like vehicles placed in a simple, controlled garage, while patients are like the same model driving on the road, subject to weather, traffic, and different drivers (akin to the immune system, tumor microenvironment, etc.). Traditional approaches focus on matching parts lists between garage and road (e.g., "both equipped with V6 engines and four tires"). While helpful, this is incomplete because identical components may fail under different conditions. DeepSADR focuses on matching functional interactions—the "circuit diagram" where ignition systems, fuel injectors, and ECUs collaborate to start the car. This functional blueprint exhibits stronger transferability between garage and real-world scenarios, even when spark plug brands differ.

Biological Terminology:

Cell Line/Patient Domain: Defined by raw genomic features ($\mathcal{G}^c, \mathcal{G}^p$). These distributions exhibit differences ($P(\mathcal{G}^c) \neq P(\mathcal{G}^p)$), resulting in distribution shifts.

Drug Response Domain: Defined by interaction patterns between drug sub-structures and gene sub-sequence. Key inhibitors often target the same apoptotic pathway—whether tested in vitro or in vivo—despite potential variations in the specific expression levels of genes within that pathway.

By modeling and integrating these functional interaction patterns (via sub-sequence interaction graphs and adaptive readout mechanisms) rather than directly processing raw genomic component lists, we bypass vast amounts of irrelevant genomic heterogeneity to precisely capture the transferable core elements of drug mechanisms of action.

## A.15 TIME REQUIRED TO CONSTRUCT THE SUBSEQUENCE INTERACTION GRAPH

Considering the need to dynamically construct separate interaction graphs for each (patient, drug) pair, and that the time required for construction depends on the size of drug and gene subsequences, we calculated the time required to construct a sub-sequence interaction graph for different drugs to visually demonstrate computational overhead. The results are shown in Table 13.

Table 13: Time consumption.

| Drug name | samples | time(ms) |
|---|---|---|
| Fluorouracil | 88 | 112 |
| Cisplatin | 40 | 61 |
| Sorafenib | 26 | 65 |
| Gemcitabine | 138 | 198 |
| Temozolomide | 46 | 82 |

## A.16 COMPARISON OF STATISTICAL SIGNIFICANCE TESTS BETWEEN DEEPSADR AND BASELINE METHODS

To demonstrate that our DeepSADR model achieves significantly better performance compared to other baseline methods, we added p-values to indicate statistical significance, with results shown in Table 14. Specifically, we computed p-values for DeepSADR versus other baseline methods across five drug outcomes under ten different random seeds. Subsequently, we combined the p-values for these five drugs using Fisher's method to obtain a more robust statistical test value. The experimental results reveal that DeepSADR achieves highly significant performance improvements over all baseline methods (all p-values $< 0.001$), a finding of substantial statistical significance. Such low p-values indicate that the observed performance gains are highly unlikely to stem from random factors or sampling errors, confirming that DeepSADR's model framework effectively enhances drug response prediction capabilities.

Table 14: Statistical Significance Test Results: Comparison of DeepSADR with Baseline Methods

| Method | AUC | | | AUPR | | |
|---|---|---|---|---|---|---|
| | Avg | Diff | p-value | Avg | Diff | p-value |
| **DeepSADR** | **0.856** | - | - | **0.862** | - | - |
| GANDALF | 0.791 | +0.064 | 5.81e-05 | 0.765 | +0.096 | 5.34e-11 |
| WISER | 0.726 | +0.129 | 2.67e-12 | 0.741 | +0.121 | 4.28e-16 |
| CODE-AE | 0.680 | +0.175 | 6.84e-12 | 0.711 | +0.151 | 4.54e-17 |
| VAEN | 0.620 | +0.235 | 2.34e-13 | 0.640 | +0.222 | 5.34e-11 |
| DAE | 0.563 | +0.293 | 2.46e-16 | 0.589 | +0.273 | 2.46e-16 |
| DruID | 0.639 | +0.217 | 3.38e-12 | 0.635 | +0.227 | 9.75e-16 |
| drug2tme | 0.634 | +0.222 | 3.97e-14 | 0.633 | +0.229 | 6.40e-15 |
| CORAL | 0.592 | +0.264 | 1.41e-16 | 0.616 | +0.245 | 1.30e-12 |
| VELODROME | 0.587 | +0.269 | 8.01e-17 | 0.539 | +0.323 | 1.44e-17 |
| CELLIGNER | 0.503 | +0.353 | 6.03e-17 | 0.526 | +0.335 | 1.44e-17 |
| DSN-DANN | 0.598 | +0.257 | 1.83e-14 | 0.621 | +0.241 | 4.91e-13 |
| DSN-MMD | 0.604 | +0.252 | 6.40e-15 | 0.622 | +0.240 | 8.55e-14 |
| TransDRP | 0.709 | +0.147 | 1.29e-10 | 0.704 | +0.158 | 2.10e-12 |
| PREDICT-AI | 0.679 | +0.176 | 1.73e-05 | 0.691 | +0.171 | 1.73e-05 |
| PANCOR | 0.652 | +0.203 | 6.40e-15 | 0.652 | +0.210 | 9.75e-16 |
| DeepTTA | 0.517 | +0.339 | 3.41e-17 | 0.544 | +0.318 | 1.44e-17 |
| GraphCDR | 0.558 | +0.297 | 1.44e-17 | 0.551 | +0.311 | 1.44e-17 |

## A.17 THEORETICAL BASIS

### A.17.1 FEATURE SPLICING STRATEGY

In transfer learning, we concatenated pre-trained features with fine-tuned features. The core idea behind this operation is to decompose drug response representations into a "general" component and a "domain-specific" component, then fuse them through concatenation to achieve better knowledge transfer.

$\hat{\mathcal{Z}}_{pre}$: Represents a universal drug response mechanism learned from extensive data on drug-cell line interactions. It encodes the most fundamental, cross-domain invariant interaction patterns between drug substructures and genomic functional pathway clusters. This feature is "frozen" and constitutes the model's prior knowledge.

$\mathcal{Z}_{fine}$: Represents patient-specific adaptations learned by the model after fine-tuning on a small dataset of patients. It captures the influence of factors unique to the patient's body but absent in cell lines—such as the immune microenvironment and tumor heterogeneity—on drug response.

**From feature decomposition to risk minimization.** As with other methods, we assume that the source domain (cell lines) and target domain (patients) share some common features, but also possess private features. However, we do not merely consider common and private features in gene expression data from cell lines and patients; in DeepSADR, common features capture domain-transcending invariant patterns (such as core mechanisms of drug-gene-substructure interactions), while private features capture domain-specific factors (e.g., patient-specific immune microenvironments or tumor heterogeneity). DeepSADR explicitly separates and fuses these two types of features by concatenating pre-trained and fine-tuned features, thereby: (1) Enhancing generalization: Regularizing the fine-tuning process with common knowledge from pre-trained features to prevent small-sample overfitting. (2) Improving predictive performance: Enabling the model to dynamically balance common knowledge and domain-specific knowledge.

Assume the true drug response $y$ is determined by both **common features** $c$ and **private features** $p$:

$$y = f(c, p) + \epsilon \tag{31}$$

where $f$ is an unknown function and $\epsilon$ represents noise. In DeepSADR: The pre-trained features $\hat{\mathcal{Z}}_{pre}$ approximate the common features $c$, learned from cell line data. The fine-tuned features $\mathcal{Z}_{fine}$ approximate the private features $p$, learned from patient data.

The concatenation operation $[\mathcal{Z}_{fine} \| \hat{\mathcal{Z}}_{pre}]$ constructs a joint feature representation, enabling the predictor $P_2$ to learn the function $f$. From the perspective of risk minimization, the expected risk is:

$$\mathcal{R}(P_2) = \mathbb{E}_{(x,y) \sim \mathcal{D}_p} \left[ \ell(P_2([\mathcal{Z}_{fine} \| \hat{\mathcal{Z}}_{pre}]), y) \right] \tag{32}$$

where $\ell$ is the loss function (MSE), and $\mathcal{D}_p$ is the patient data distribution. By minimizing this risk, the model simultaneously utilizes:

- **Common Feature Stability**: $\hat{\mathcal{Z}}_{pre}$ is learned from large-scale cell line data, providing high-confidence prior knowledge.
- **Private Feature Adaptability**: $\mathcal{Z}_{fine}$ captures patient-specific mechanisms through fine-tuning.

This decomposition effectively reduces overfitting risk in small-sample learning because:

- When patient data is scarce, the model tends to rely on the common features in $\hat{\mathcal{Z}}_{pre}$, avoiding overfitting to noise.
- The fine-tuning process only updates a small number of parameters (AR and $P_2$), further constraining model complexity.

Table 15: Results of drug cold experiments conducted on four different drugs

| Methods | Doxorubicin | | Erlotinib | | Oxaliplatin | | Sunitinib | |
|---|---|---|---|---|---|---|---|---|
| | AUC↑ | AUPR↑ | AUC↑ | AUPR↑ | AUC↑ | AUPR↑ | AUC↑ | AUPR↑ |
| **DeepSADR** | **0.855/0.075** | **0.848/0.052** | **0.883/0.044** | **0.881/0.049** | **0.892/0.048** | **0.884/0.045** | **0.863/0.061** | **0.871/0.056** |
| WISER | 0.795/0.056 | 0.811/0.093 | 0.807/0.106 | 0.826/0.091 | 0.751/0.091 | 0.752/0.084 | 0.797/0.087 | 0.783/0.092 |
| GANDALF | 0.799/0.077 | 0.808/0.098 | 0.801/0.074 | 0.798/0.081 | 0.751/0.071 | 0.749/0.094 | 0.749/0.089 | 0.737/0.096 |
| CODE-AE | 0.794/0.091 | 0.789/0.076 | 0.801/0.071 | 0.702/0.089 | 0.771/0.082 | 0.785/0.092 | 0.747/0.096 | 0.758/0.106 |
| TransDRP | 0.734/0.057 | 0.755/0.101 | 0.788/0.058 | 0.782/0.062 | 0.749/0.089 | 0.751/0.101 | 0.763/0.089 | 0.784/0.103 |

*Note:* Data related to clinical relapse is used for all evaluations. The results are reported as the mean/standard deviation of multiple random seeds. Best performer among all baselines is in **bold**.

## A.18 DRUG COLD SETTING

To further validate our model's predictive capability for drugs not yet tested in human trials, we conducted a cold-start experiment. We selected four drugs from the patient drug response dataset that were not used during fine-tuning as untested drugs, and applied the fine-tuned model to predict outcomes for these four drugs. The corresponding results are shown in Table 15.

## A.19 OTHER EXPERIMENT

Table 16: Performance (AUC and AUPR scores) for 5 clinical drugs

| Methods | Fluorouracil | | Temozolomide | | Sorafenib | | Gemcitabine | | Cisplatin | |
|---|---|---|---|---|---|---|---|---|---|---|
| | AUC↑ | AUPR↑ | AUC↑ | AUPR↑ | AUC↑ | AUPR↑ | AUC↑ | AUPR↑ | AUC↑ | AUPR↑ |
| **DeepSADR** | **0.805/0.056** | **0.821/0.023** | **0.870/0.026** | **0.886/0.029** | **0.957/0.037** | **0.978/0.024** | **0.719/0.057** | **0.702/0.022** | **0.927/0.027** | **0.922/0.021** |
| DeepSADR - | 0.801/0.133 | 0.817/0.107 | 0.861/0.122 | 0.879/0.131 | 0.951/0.142 | 0.970/0.104 | 0.716/0.078 | 0.701/0.092 | 0.919/0.102 | 0.921/0.097 |
| DeepSADR * | 0.796/0.103 | 0.803/0.067 | 0.821/0.092 | 0.809/0.093 | 0.917/0.074 | 0.907/0.084 | 0.719/0.057 | 0.702/0.022 | 0.897/0.077 | 0.892/0.078 |

*Note:* Data related to clinical relapse is used for all evaluations. The results are reported as the mean/standard deviation of multiple random seeds. Best performer among all baselines is in **bold**.'*' Indicates that only the top 30% of association weights are used in the sub-sequence interaction subgraph to replace threshold selection. '-' denotes results obtained by training the model using standard cross-entropy loss.

## A.20 GENE PARTITIONING STRATEGY

To achieve pathway clustering of genes, we designed a method called Enrichment-based Gene Functional Clustering (ENRICH) within the DeepSADR framework (Kim et al., 2008). This method combines prior biological knowledge with data-driven pattern discovery to specify gene clusters, and each pathway cluster coherent module may include multiple biological pathways that exhibit synergistic roles in cellular processes. The specific implementation process is as follows:

- **Multi-source pathway knowledge integration**: First, we systematically extracted functional pathway association information for 1,426 genes from authoritative biological pathway databases such as KEGG 2021 Human(Kanehisa et al., 2021) and Gene Ontology Biological Process 2021(Consortium et al., 2021) within the patient drug response database. This enables the construction of a comprehensive pathway-gene mapping network, providing a biological foundation for subsequent functional module clustering.

- **Manual Definition of Core Functional Modules**: Based on key mechanisms in cancer biology, we predefined a set of core functional categories (e.g., signal transduction, cell cycle regulation, DNA damage repair, immune response, and metabolic pathways). Each category uses pathway keyword matching algorithms to preliminarily classify relevant genes into corresponding modules, with upper and lower limits set for module size to ensure practicality and Flexibility.

- **Adaptive Clustering for Unassigned Genes**: For genes not assigned via predefined rules, hierarchical clustering was applied to intelligently regroup them based on co-occurrence patterns across multiple pathways. The number of clusters dynamically adjusted according

> to the size of remaining genes and total target modules, forming "automatically discovered modules" that effectively capture novel functional combinations hidden in the data.

- **Full Coverage Assurance Mechanism**: We introduced a mandatory assignment verification step. By applying pathway similarity matching and prioritizing the smallest modules, we guarantee 100

- **Multi-dimensional Quality Validation System**: Each generated module undergoes quality assessment via functional enrichment analysis (evaluating significantly enriched biological pathways) and the Jaccard similarity index (measuring functional consistency between the module and the global background). This ensures each module possesses both functional specificity and biological coherence.

In this paper, the number of gene functional clusters was determined through multiple experimental validations, it serves as a flexible and adjustable parameter. We conducted experimental verifications with various cluster sizes, with the specific results presented in Table 17.

Table 17: Experimental results for different number of gene functional clusters(average of 5 drugs).

| Methods | AUC↑ | AUPR↑ |
|---|---|---|
| DeepSADR | **0.856** | **0.862** |
| DeepSADR[1] | 0.762 | 0.775 |
| DeepSADR[6] | 0.781 | 0.779 |
| DeepSADR[11] | 0.847 | 0.854 |
| DeepSADR[12] | 0.849 | 0.855 |
| DeepSADR[14] | 0.848 | 0.859 |
| DeepSADR[15] | 0.846 | 0.858 |
| DeepSADR[80] | 0.763 | 0.751 |
| DeepSADR[1426] | 0.653 | 0.639 |

The superscript in the first column of the table denotes the number of gene functional clusters set for DeepSADR. DeepSADR represents the optimal model from our paper, with a functional clusters count set to 13.

DeepSADR[1] indicates that we do not cluster genes according to the algorithm, treating all genes as a unified whole. DeepSADR[1426] indicates that each gene as a functional cluster

The results of the DeepSADR[1] experiment demonstrate that our functional clustering and grouping of genes is effective. The results of DeepSADR[6] indicate that an insufficient number of clusters leads to "underfitting" in the model, preventing it from capturing more complex underlying patterns in the data related to drug and gene functional clustering. Conversely, an excessive number of clusters(DeepSADR[80] and DeepSADR[1426] ) causes the model to "overfit", causing it to rigidly memorize noise and idiosyncrasies from a small number of samples rather than learning generalizable patterns. Additionally, we observed that experimental results remained relatively stable with minimal variation when the number of clusters was set between 11 and 15. After comprehensive consideration, we selected 13 as the optimal value for this study.

Table 18 details the 13 gene functional clusters.

Table 18: The introduction of 13 Gene Functional Clusters.

| ID | Gene functional Cluster | introduction |
|----|------------------------|--------------|
| 1 | metabolic | Fundamental cellular life processes including energy metabolism, nutrient synthesis and degradation, and small molecule metabolism. |
| 2 | signal pathway | The recognition, transduction, and regulation of signals such as hormones, growth factors, and cytokines involve cell proliferation, differentiation, and stress responses. |
| 3 | Cell growth and differentiation | Stem cell differentiation, tissue development, neurogenesis, and other developmental and regenerative processes. |
| 4 | Cell cycle and proliferation | DNA replication, mitosis, initiation and termination of cell proliferation. |
| 5 | Immune-related | Innate and Adaptive Immunity, Inflammatory Response, Pathogen Recognition and Clearance. |
| 6 | Cell transport and localization | Protein transport, endocytosis, ion transport across membranes, etc. |
| 7 | Cell stress and apoptosis | Oxidative stress, hypoxia, toxic substance responses, and programmed cell death (apoptosis) regulation. |
| 8 | Cytoskeleton and cell motor | Cytoskeletal reorganization, cell migration, and shape maintenance. |
| 9 | Disease-associated | Mechanisms directly associated with specific diseases (such as cancer, neurodegenerative diseases, infectious diseases). |
| 10 | Other | Sensory perception, tissue repair, gene expression regulation, ion balance, and other diverse life activities. |
| 11 | Cancer | Mechanisms associated with cancer initiation and progression, such as unlimited proliferation, evasion of apoptosis, and angiogenesis. |
| 12 | Gene expression regulation | Transcription regulation, RNA processing, epigenetic regulation, etc. |
| 13 | Unknow | Pathways lacking clear classification or annotation information may represent novel biological processes or functions not yet fully annotated in databases. |

