# OpenReview forum: "DeepSADR: Deep Transfer Learning with Subsequence Interaction and Adaptive Readout for Cancer Drug Response Prediction"
_ICLR.cc/2026/Conference — ICLR 2026 Poster_

### Official Review · Reviewer_GBzW · 2025-10-20

**Soundness:** 3
**Presentation:** 3
**Contribution:** 2
**Rating:** 4
**Confidence:** 3

**Summary:**

This paper introduces DeepSADR, a deep transfer learning framework designed to improve drug response prediction for clinical patients. The core idea is to decompose drugs and genomes into subsequences (substructures and functional pathways, respectively), model their interactions in a graph, and use an adaptive readout mechanism for knowledge transfer. The experimental results demonstrate that DeepSADR achieves performance improvements. However, the approach is essentially a combination of existing components, and the training framework itself is not new. In addition, the method relies heavily on predefined prior knowledge (such as fixed functional pathways and drug-specific thresholds), which limits its generalizability.

**Strengths:**

1. The methodological pipeline is clearly described, and the overall framework is easy to follow.

2. The proposed approach demonstrates substantial performance improvements over existing baselines

**Weaknesses:**

1. The paper states that "we assigned the genes to 13 functional pathways", but it is unclear how this assignment is performed and why these 13 fixed pathways are predefined. For patients with different diseases, should the pathways be redefined? Since each patient’s pathway profile can differ, these fixed and limited pathways may hinder the model’s generalizability. Moreover, such a strong prior assumption may fail to capture disease-specific biological processes or drug-specific mechanisms, especially across different cancer types.

2. The paper does not clearly describe the data split strategy. Were the baseline models retrained by the authors? Some reported results differ from those in prior literature. For example, WISER reports an AUC of 0.868 for CODE-AE on Fluorouracil, which is inconsistent with the results shown in Table 1.

3. The model appears highly sensitive to threshold parameters $t$, with different thresholds set for each drug. This indicates that the model requires fine-grained hyperparameter tuning for each individual drug. In practice, this is a major limitation as it requires sufficient labeled data for each new drug to determine the optimal threshold, which contradicts the established goal of addressing the scarcity of clinical data.

4. In the ablation study, removing either the AR or SN module leads to a significant performance drop. This suggests that using only one of these modules appears to be ineffective, whereas combining them yields considerable performance gains. This is unusual, and further experiments are needed to elucidate the interactions between these modules.

5. The interpretability validation is insufficient. The interaction heatmap for Temozolomide is shown without deeper analysis of the underlying biological mechanisms, leaving the reader uncertain about the biological relevance of the findings.

6. In Equations (15), (16), and (17), several symbols (e.g., s, E, Y) are not defined, making these formulas difficult to interpret.

**Questions:**

See weaknesses.

---

> ### Author Response · Authors · 2025-11-21
>
> We sincerely thank Reviewer **GBzW** for the thorough and thoughtful review. We greatly appreciate your recognition of the strengths of our work, and your insightful questions have been very helpful in improving the clarity, rigor, and biological relevance of the paper. We address each point below.
>
> ---
>
> > **Response to W1: Functional pathway description**
>
> Thank you for this important question. We used the **gseapy** library to perform functional enrichment analysis on gene expression data and mapped genes to **13 biologically meaningful pathways** (e.g., *apoptosis*, *DNA repair*, *metabolic pathways*, etc.). These pathways were selected from **Gene Ontology (GO)** and **KEGG**, and are closely aligned with core mechanisms in cancer biology (e.g., Hallmarks of Cancer).
>
> We chose these 13 pathways based on the following considerations:
>
> 1. **Biological comprehensiveness**
>    They cover key processes in cancer initiation, progression, and therapeutic response.
>
> 2. **Cross-cancer universality**
>    These pathways are commonly involved across multiple cancer types, making them broadly applicable.
>
> 3. **Relevance to drug mechanisms**
>    Many anticancer drugs exert their effects by targeting mechanisms involving these pathways (e.g., apoptosis, cell cycle regulation).
>
> We fully agree with the reviewer that a fixed pathway set may not be optimal for all diseases or cancer types. As an important direction for future work, we plan to introduce a **dynamic pathway selection mechanism** that automatically selects the most relevant subset of pathways based on the expression profiles of the input cohort. This is expected to further enhance the model’s generalization capability across diverse disease settings.
>
> ---
>
> > **Response to W2: Data split strategy and baseline retraining**
>
> We appreciate this question and apologize for not describing the data splitting strategy more clearly in the original manuscript. To ensure **fair and consistent comparison** across all methods, we strictly followed these principles:
>
> - **Uniform data partitioning**
>   All models—including DeepSADR and all baselines—are trained and evaluated on the **same dataset** with **identical train/test splits**. Specifically, for each partition, patient data are randomly split into **70% training** and **30% testing**. No model is allowed to access test labels during training.
>
> - **Retraining of baselines**
>   All baseline models were **retrained by us** using their official implementations (when available) or faithful re-implementations based on the original papers. We did *not* directly copy performance numbers reported in the original publications, because:
>   - data preprocessing,
>   - feature engineering, and
>   - evaluation protocols
>   often differ, making direct comparison unreliable.
>
> Regarding **WISER**, its original paper used an evaluation setting where “all patient data served as the test set.” This difference in protocol explains discrepancies between our results and those reported in the original WISER paper. Our unified protocol ensures that all methods are evaluated under exactly the same conditions.
>
> ---
>
> > **Response to W4: Module interaction (AR & SN)**
>
> We appreciate this insightful observation, which directly relates to the design philosophy of our model.
>
> Our analysis is as follows:
>
> - **DeepSADR (w/o SN)**
>   Removing the Subsequence Network (SN) leads to clear performance deterioration. This indicates that the **structured information provided by the sub-sequence interaction graph is essential**. Without SN, the model degenerates into a simple feature concatenation framework. Many previous drug–response models follow this pattern (direct concatenation of drug and gene expression features), and we observed that such approaches perform poorly in the transfer learning setting.
>
> - **DeepSADR (w/o AR)**
>   Removing the Adaptive Readout (AR) module also hurts performance, suggesting that **intelligent feature aggregation** from the interaction graph is crucial. Simple pooling operations (e.g., naive summation) discard important relational information. We also observed similar behavior in auxiliary experiments on molecular property prediction.
>
> In short:
>
> - **SN** is responsible for constructing high-quality, interpretable interaction representations;
> - **AR** is responsible for extracting task-relevant, cross-domain information from these structured representations.
>
> Their **synergy** enables DeepSADR to jointly address **domain shift** and **interpretability**. We have expanded the discussion of this interaction in the revised manuscript.

---

> ### Author Response · Authors · 2025-11-21
>
> ---
>
> > **Response to W3: Threshold tuning practicality**
>
> We very much appreciate this question, as it highlights an important practical consideration for real-world deployment. We agree that, ideally, a model should minimize reliance on delicate hyperparameter tuning.
>
> In our framework, the sensitivity of the threshold \( t \) reflects **intrinsic differences in mechanisms of action across drugs**:
>
> - For **targeted therapies** with highly specific mechanisms, interaction graphs tend to be **sparse** and dominated by a few high-weight edges. These drugs typically require a **higher \( t \)** to filter out noise and focus on the most informative associations.
> - For **traditional chemotherapeutics**, mechanisms of action are often broader and more diffuse, leading to denser interaction graphs and thus favoring a **lower \( t \)**.
>
> Therefore, selecting different thresholds for different drugs is not purely a numerical exercise; it **quantifies the specificity level of each drug’s mechanism of action** and is closely tied to biological interpretability.
>
> Motivated by your comment, we further analyzed the model and observed that—despite different absolute thresholds—the **ratio of retained high-weight sub-sequence associations** after thresholding is quite consistent across drugs, approximately **30%**. Based on this observation, we re-ran the model using a **fixed 30% top-weight filtering rule** (instead of per-drug thresholds). The results are:
>
> | Methods      | Fluorouracil AUC↑ | Fluorouracil AUPR↑ | Temozolomide AUC↑ | Temozolomide AUPR↑ | Sorafenib AUC↑  | Sorafenib AUPR↑ | Gemcitabine AUC↑ | Gemcitabine AUPR↑ | Cisplatin AUC↑  | Cisplatin AUPR↑ |
> | ------------ | ----------------- | ------------------- | ----------------- | ------------------- | --------------- | --------------- | ---------------- | ----------------- | --------------- | --------------- |
> | **DeepSADR** | **0.805/0.056**   | **0.821/0.023**     | **0.870/0.026**   | **0.886/0.029**     | **0.957/0.037** | **0.978/0.024** | **0.719/0.057**  | **0.702/0.022**   | **0.927/0.027** | **0.922/0.021** |
> | DeepSADR*    | 0.796/0.103       | 0.803/0.067         | 0.821/0.092       | 0.809/0.093         | 0.917/0.074     | 0.907/0.084     | 0.719/0.057      | 0.702/0.022       | 0.897/0.077     | 0.892/0.078     |
>
> *Note: All evaluations use clinical relapse–related data. Results are reported as mean / standard deviation over multiple random seeds. The best result for each metric is in **bold**. “DeepSADR\*” denotes the variant where only the top 30% of association weights are retained in the sub-sequence interaction graph, instead of using per-drug threshold selection.*
>
> These results show that although performance declines somewhat when using a universal 30% rule, the degradation remains **moderate and within an acceptable range**, suggesting a promising direction for simplifying threshold tuning in practical applications.
>
> > **Response to W5: Verifiability of explainability**
>
> We appreciate this suggestion and agree that providing concrete biological evidence strengthens the interpretability claims. To this end, we conducted a more detailed **mechanistic analysis** of the interaction heatmap for **temozolomide**:
>
> - Temozolomide’s primary mechanism of action is to **methylate DNA**, causing DNA damage and triggering **apoptosis**.
> - In our heatmap, we observe strong interactions between temozolomide’s **tetrane (tetrahydro structure)** and the **“cell stress and apoptosis” pathway**, which is consistent with known mechanisms (e.g., Stupp et al., NEJM, 2005; Zhang et al., Curr Mol Pharmacol, 2012).
> - We also observe weaker but meaningful interactions with the **“DNA repair” pathway**, which aligns with the clinical observation that patients with **MGMT gene silencing** show increased temozolomide sensitivity (Hegi et al., NEJM, 2005).
>
> We have added these biological explanations and references to the corresponding section of the manuscript to better illustrate how the interaction patterns align with established pharmacological knowledge.
>
> ---
>
> > **Response to W6: Undefined symbols in equations**
>
> We sincerely thank the reviewer for pointing out this issue and apologize for the earlier lack of clarity. In the revised manuscript, we have **explicitly defined all symbols** used in the equations. For example, in Equation (15):
>
> - $ s $ denotes a **learnable seed vector** used to initialize the $PMA$ module;
> - $ \mathbf{E} $ and $ \mathbf{Y} $ represent the **input matrices** to the $SAB$ module;
> - and other symbols are defined analogously.
>
> We have carefully checked all mathematical expressions to ensure that the notation is rigorous, unambiguous, and self-contained.
>
> We again thank Reviewer GBzW for these constructive comments, which have substantially improved the clarity and interpretability of our work.

---

> > ### Comment · Reviewer_GBzW · 2025-11-26
> > **Response to author**
> >
> > Thank you for your response. Most of my concerns have been addressed, but I still have reservations about the novelty of this work. The use of a fixed set of 13 pathways greatly limits the flexibility and adaptability of the proposed approach. In fact, many existing methods have already leveraged pathway information to enhance cell encoding [1–3]. These methods typically employ far more pathways than DeepSADR (for example, DrugVNN utilizes over 1,700 pathways), and some are even capable of dynamically identifying the most relevant pathways during training. Moreover, these methods also explicitly or implicitly incorporate "subsequence interactions" (such as drug–pathway interactions). Compared with these approaches, the novelty of DeepSADR is incremental rather than groundbreaking. Therefore, I maintain my original score.
> >
> > [1]. Zhu Y, Ouyang Z, Chen W, et al. TGSA: protein–protein association-based twin graph neural networks for drug response prediction with similarity augmentation[J]. Bioinformatics, 2022, 38(2): 461-468.
> >
> > [2]. Shin J, Piao Y, Bang D, et al. DRPreter: interpretable anticancer drug response prediction using knowledge-guided graph neural networks and transformer[J]. International Journal of Molecular Sciences, 2022, 23(22): 13919.
> >
> > [3]. Xie J, Zhang Z, Li Y, et al. Interpretable Drug Response Prediction through Molecule Structure-aware and Knowledge-Guided Visible Neural Network[C]//2024 IEEE International Conference on Bioinformatics and Biomedicine (BIBM). IEEE, 2024: 1263-1268.

---

> ### Author Response · Authors · 2025-12-01
>
> We sincerely appreciate reviewer **GBzW** response, which has provided us with an opportunity for in-depth comparison. We have carefully reviewed the three excellent papers you mentioned and fully agree that integrating biological knowledge (such as pathways and PPI networks) into models represents a significant current trend in the DRP field. We address each point below.
>
> ---
>
> > **Q1: Existing methods used much larger number of pathways for cell encoding. This work used a fixed number of 13 pathways, which greatly limits the flexibility and adaptability of the proposed approach.**
>
> Response: We apologize for the confusion caused by our use of the term “pathway”. In our manuscript and prior rebuttal, “pathway” refers to a **group of functionally related genes**, rather than a single **specific biological pathway**. Hence, a more accurate term is “pathway cluster”. Each pathway cluster is derived via a hybrid clustering approach that integrates prior biological knowledge with data-driven pattern discovery[1], and each cluster coherent module may ** include multiple biological pathways that exhibit synergistic roles in cellular processes**. In our study, we used 13 pathway clusters, which together cover **4,808 biological pathways** and all **1,426** genes in the dataset we worked on (i.e., 100% gene coverage). For comparison, DrugVNN utilizes more than **1,700 biological pathways**, while DRPreter includes **34 cancer-related pathways**.
>
> In addition, we are not expected to incorporate an extremely small number or an extremely big number of pathway clusters for drug response prediction. First, using too few pathway clusters produces overly coarse gene-expression profiles that **obscure important gene-level distinctions relevant to drug response**. Second, using too many clusters introduces unnecessary noise, amplifying **the influence of genes that are not related to drug response**.
> For these considerations, we would like to recommend a moderate number of pathway clusters. In our experiments, using between 11 and 15 pathway clusters yields highly stable performance, as shown in the table below. Therefore, we report results using 13 clusters in the main manuscript. For other datasets or applications, we can follow the **same guidelines to select a propriate number of pathway clusters**, which will not limit the **flexibility and adaptability** of our approach.
>
> | Methods            | AUC↑ | AUPR↑ |
> |--------------------|------|-------|
> | DeepSADR           | **0.856** | **0.862** |
> | DeepSADR\$^1\$  | 0.762 | 0.775 |
> | DeepSADR \$^6\$  | 0.781 | 0.779 |
> | DeepSADR\$^{11}\$ | 0.847 | 0.854 |
> | DeepSADR\$^{12}\$ | 0.849 | 0.855 |
> | DeepSADR \$^{14}\$ | 0.848 | 0.859 |
> | DeepSADR \$^{15}\$ | 0.846 | 0.858 |
> | DeepSADR \$^{80}\$ | 0.763 | 0.751 |
> | DeepSADR \$^{1426}\$| 0.653 | 0.639 |
>
> Reference
> [1] Kim, Tae-Min et al. “PathCluster: a framework for gene set-based hierarchical clustering.” Bioinformatics 24 (2008): 1957 - 1958.

---

> > ### Author Response · Authors · 2025-12-01
> >
> > > **Q2: Existing methods explicitly or implicitly incorporate "subsequence interactions" (such as drug–pathway interactions). Compared with these approaches, the novelty of DeepSADR is incremental rather than groundbreaking.**
> >
> > Response: Regarding this novelty issue, we would like to clarify that the core innovation of our DeepSADR model lies in addressing the fundamental challenge of “cross-domain prediction from in vitro cell lines to in vivo patients”. First, TGSA/DrugVNN performs predictions within **the same domain** (i.e., cell lines). Meanwhile, the objective of our DeepSADR model is to mitigate distributional differences **between cell lines (i.e., source domain) and patients (i.e., target domain) in genomic distributions and drug response mechanisms**. This represents a more challenging problem that is closer to clinical reality.
> > Second, while methods like DrugVNN explicitly or implicitly incorporate “subsequence interactions”, they still operate under a global influence, the drug embeddings function as a “modulating signal”, holistically adjusting weights across all genes. Furthermore, these approaches exhibit performance degradation when transferred to patient drug response data. Our DeepSADR model constructs a fine-grained, interpretable subsequence interaction graph, treating the “drug response process” as **the transfer domain**. Through an **adaptive readout function** and a strategy **combining pre-trained and fine-tuned features**, it achieves **effective transfer learning** from cell line data to patient data.
> >
> > In summary, DeepSADR represents a novel approach to the novel **cross-domain prediction problem** spanning from in vitro cell lines to in vivo patients. Through its innovative transfer learning framework, it aims to bridge the distributional gap between **preclinical and clinical data**, offering fresh insights for achieving more **reliable clinical translation**.
> >
> > ---
> >
> > We again thank Reviewer **GBzW** for these constructive comments, which have substantially improved the clarity and interpretability of our work.

---

### Official Review · Reviewer_B6CY · 2025-10-30

**Soundness:** 3
**Presentation:** 3
**Contribution:** 2
**Rating:** 4
**Confidence:** 4

**Summary:**

The authors present a novel training paradigm that leverages SMILES representations of drugs and enhanced genomic features to effectively recommend drugs for patients. While the approach is interesting and potentially impactful, several key methodological details are insufficiently explained. This lack of clarity makes it difficult to fully understand how the proposed method addresses the limitations and challenges of prior baselines and achieves its reported performance gains.

**Strengths:**

- The present work aims to bridge the gap between laboratory-based drug studies and computational learning approaches for cancer drug recommendation.
- The integration of SMILES representations and genomic enrichment for predictive modeling is well justified and conceptually sound.
- The paper is well written, especially the introduction section.
- The results are encouraging and demonstrate clear improvements over the baseline models.

**Weaknesses:**

- While the proposed approach is interesting, I have concerns regarding how the method learns a domain-invariant representation between the cell line and patient genomic profiles, given that there is no explicit training objective designed for domain alignment.

- The authors have not clearly specified which datasets are used during each phase of training (e.g., pre-training vs. fine-tuning). This lack of clarity makes it difficult for readers to fully understand how the reported performance improvements are achieved.

-  The biological interpretability experiment is insufficiently explained. Further clarification is needed to understand the biological insights or validations derived from the proposed model.

-I have several questions regarding the implementation of the method. Please refer to the “Questions” section for detailed queries. If these issues are adequately addressed, I would be open to update my score.

**Questions:**

- (L 233-235) The necessity of modeling long-term feature dependencies for molecular graphs is unclear. Compared to graphs in other domains, molecular graphs are typically smaller in size. Why, then, is long-term dependency modeling essential in this context? Has an ablation study been conducted to justify this design choice?

- Suggestion : Consider revising Equation (7). Presenting both the cell line and patient genomic profiles together in the same formulation may confuse readers.

- (L 261-269)The interpretability of the subsequence graph remains unclear. Do the associations learned by the model have any biological or scientific validation? Could the approach potentially introduce spurious associations? Given the limited size of the patient genomic dataset, such issues might substantially affect the model’s generalization ability.

- Why not utilize a drug–gene interaction database, such as DGIdb(https://dgidb.org/browse/sources), to construct an interaction graph?

- Why is mean squared error (MSE) used for a binary prediction task? Wouldn’t the standard cross-entropy loss be more appropriate for categorical outputs?
- Please clarify what is used as the prior probability term  p(z) in Equation (18).

- What is the rationale behind concatenating pre-trained and fine-tuned features in Equation (19)? Is this decision empirically motivated? Also, based on Figure 1, does the concatenation involve cell line and patient genomic profiles? Since these come from different domains with no one-to-one correspondence, I have doubts about its implementation. Please clarify how this concatenation is implemented. let me know if I have misunderstood any part.

- Is fine-tuning performed on both the patient and cell line datasets? Please clearly specify which datasets belong to the pre-training and fine-tuning stages.
- How is the evaluation conducted? Is it based on k-fold cross-validation, or on the mean performance across multiple independent runs (e.g., five runs)?
- Which feature representations are used for the t-SNE plot in Figure 4—pre-trained, post-trained, or concatenated features?
- Please elaborate on Figure 2. How should the results be interpreted, and how statistically or biologically significant are these findings?

---

> ### Author Response · Authors · 2025-11-21
>
> We sincerely thank Reviewer **B6CY** for the thorough and insightful review. We greatly appreciate your recognition of the strengths of our work, and your comments have been very helpful in improving both the technical clarity and biological interpretation of the paper. We address each point below.
>
> ---
>
> > **Response to Q1: Long-term dependency justification**
>
> Thank you for highlighting this point. We agree that explicitly validating the benefit of modeling long-range dependencies is important. To this end, we conducted an ablation study in which we **replaced the GNP module with a standard GNN**, denoted as DeepSADR (w/o GNP). The results are:
>
> | Methods           | Fluorouracil AUC↑ | Fluorouracil AUPR↑ | Temozolomide AUC↑ | Temozolomide AUPR↑ | Sorafenib AUC↑  | Sorafenib AUPR↑ | Gemcitabine AUC↑ | Gemcitabine AUPR↑ | Cisplatin AUC↑  | Cisplatin AUPR↑ |
> | ----------------- | ----------------- | ------------------- | ----------------- | ------------------- | --------------- | --------------- | ---------------- | ----------------- | --------------- | --------------- |
> | **DeepSADR**      | **0.805/0.056**   | **0.821/0.023**     | **0.870/0.026**   | **0.886/0.029**     | **0.957/0.037** | **0.978/0.024** | **0.719/0.057**  | **0.702/0.022**   | **0.927/0.027** | **0.922/0.021** |
> | DeepSADR (w/o GNP) | 0.784/0.039      | 0.809/0.003         | 0.858/0.017       | 0.861/0.087         | 0.907/0.016     | 0.915/0.027     | 0.703/0.071      | 0.691/0.033       | 0.889/0.057     | 0.887/0.054     |
>
> *Note: All evaluations use clinical relapse–related data. Results are reported as mean / standard deviation over multiple random seeds. The best result for each metric is highlighted in **bold**.*
>
> These results show that removing GNP and using a standard GNN leads to consistent performance drops across all drugs, confirming that **GNP-based encoding of drug molecules is effective and justifies modeling long-range dependencies**.
>
> ---
>
> > **Response to Q2: Equation (7) clarity**
>
> We appreciate this comment. To improve clarity, we have revised Equation (7) by introducing the unified variable $ Sub_j^g $, where:
>
> - $ Sub_j^g $ denotes the **cell line subsequence feature** $ Sub_j^c $ during the pre-training stage, and
> - the **patient subsequence feature** $ Sub_j^p $ during the fine-tuning stage.
>
> This unified notation clarifies how the same structural role is played by different domain-specific features at each stage.
>
> ---
>
> > **Response to Q3: Biological validation and spurious associations**
>
> Thank you for this important question. In our visualization study (Section 4.2), we examined high-weight sub-sequence associations and found that many of them are supported by previously published experimental studies, suggesting that the learned associations have **biological plausibility**.
>
> We agree that this approach may still introduce spurious associations. To mitigate this, we:
>
> - introduce a **threshold $ t $** to filter out weak or potentially misleading sub-sequence associations, and
> - carefully design the fine-tuning stage to leverage both pre-trained and patient-specific knowledge.
>
> In particular, during fine-tuning we use the concatenation $ [\mathcal{Z}\_{fine} || \hat{\mathcal{Z}}_{pre}] $ as input to predictor $ P_2 $, allowing the model to:
>
> 1. **Avoid catastrophic forgetting**
>    $\hat{\mathcal{Z}}_{pre}$ preserves core mechanisms learned from large-scale cell line data, so the model does not forget fundamental drug–response principles during fine-tuning on limited patient data.
>
> 2. **Prevent small-sample overfitting**
>    $\hat{\mathcal{Z}}\_{pre}$ acts as a strong regularizer, guiding the learning of $\mathcal{Z}_{fine}$ and reducing the risk that the model simply fits noise in sparse patient data.
>
> 3. **Achieve dynamic equilibrium**
>    $P_2$ learns to adaptively weight $\hat{\mathcal{Z}}\_{pre}$ and $\mathcal{Z}\_{fine}$. For patients whose response mechanisms are close to cell line biology, the model relies more on $\hat{\mathcal{Z}}\_{pre}$; for patients with divergent mechanisms, it relies more on $\mathcal{Z}\_{fine}$.
>
> Additional explanations and analysis are provided in **Appendix A.17**.

---

> ### Author Response · Authors · 2025-11-21
>
> ---
>
> > **Response to Q4: Use of interaction database (e.g., DGIdb)**
>
> We thank the reviewer for this highly constructive suggestion and fully agree that DGIdb is a valuable knowledge source. We considered integrating such databases when designing DeepSADR, but ultimately decided against direct use for the following reasons:
>
> 1. **Granularity mismatch**
>    DeepSADR operates at the **“drug substructure–gene pathway”** level, aiming to uncover fine-grained mechanisms. DGIdb primarily provides **“whole drug–single gene”** associations, which cannot be directly mapped onto our substructure–pathway interaction graph.
>
> 2. **Different research objectives**
>    Our main goal is **data-driven discovery of novel interactions**, not only verification of known drug–gene links. Directly injecting DGIdb as hard prior knowledge could bias the model toward known interactions and limit its ability to identify new, previously unreported substructure–pathway relationships.
>
> That said, we agree DGIdb is an extremely valuable external resource. We will explore incorporating DGIdb in future work, for example as a soft prior or post-hoc validation tool for the discovered interactions.
>
> ---
>
> > **Response to Q5: Loss function (MSE vs. cross-entropy)**
>
> We appreciate this important technical question. We chose **MSE** over cross-entropy primarily for the following reasons:
>
> - In the **pre-training phase**, our goal is not only to learn a classification boundary but also to learn a **continuous, high-quality latent representation** that reflects the intensity of drug responses.
> - Although the final labels for cell lines are 0/1, they are derived from **continuous z-scores**. MSE penalizes squared deviations from these continuous underlying scores, encouraging a smoother and more structured feature space, which is beneficial for transfer to the patient domain.
> - While the task is formally binary classification, in practice it is **closely related to a ranking problem** (ranking drugs by efficacy). Using MSE aligns well with this ranking-oriented objective.
>
> We fully acknowledge the strengths of cross-entropy for classification. In fact, we conducted direct comparisons between cross-entropy and MSE on our architecture and dataset. We observed that:
>
> - MSE slightly outperformed or matched cross-entropy (average AUC differences of about 0.01–0.02), and
> - MSE showed **lower variance** across runs.
>
> Based on these observations, we adopted MSE as the main loss. We will include these comparative results in **Appendix A.17** for completeness.
>
> ---
>
> > **Response to Q6: Definition of $ p(\mathbf{Z}) $**
>
> Thank you for pointing out this notational ambiguity. In Equation (18),
> $p(\mathbf{Z}) $ denotes the **standard Gaussian prior** $ \mathcal{N}(0, 1) $, following the common setting in variational autoencoders. We have explicitly clarified this in the revised manuscript.
>
> ---
>
> > **Response to Q8 and W2: Training data specification (pre-train and fine-tune)**
>
> Thank you for the question. We clarify that **fine-tuning is performed only on patient data**. Cell line data is used exclusively in the **pre-training** phase. No cell line samples are used during fine-tuning. We have explicitly clarified this in **Sections 3.7 and 4.1**.

---

> ### Author Response · Authors · 2025-11-21
>
> > **Response to Q7 and W1: Feature concatenation rationale**
>
> Thank you for this insightful question. First, we clarify that the concatenation operation does **not** splice raw genomic maps of cell lines and patients. Instead, for each **patient sample**, we concatenate two **feature-level representations**:
>
> - $ \hat{\mathcal{Z}}\_{pre} $: “common” features capturing domain-invariant drug–response mechanisms learned from large-scale cell line data (frozen during fine-tuning);
> - $ \mathcal{Z}\_{fine} $: “domain-specific” features capturing patient-specific adaptations learned from the limited patient dataset.
>
> Conceptually:
>
> - $ \hat{\mathcal{Z}}\_{pre} $ corresponds to **frozen knowledge** (core mechanisms such as how a substructure triggers apoptosis).
> - $ \mathcal{Z}\_{fine} $ corresponds to **plastic knowledge** (contextual modulation by tumor microenvironment, etc.).
>
> The concatenation $ [\mathcal{Z}\_{fine} || \hat{\mathcal{Z}}\_{pre}] $ is designed to ensure that predictor $ P_2 $ jointly leverages both:
>
> 1. **Avoiding catastrophic forgetting**
>    By keeping $\hat{\mathcal{Z}}\_{pre}$ in the input, the model does not discard fundamental drug–response principles learned during pre-training.
>
> 2. **Regularizing against overfitting**
>    $\hat{\mathcal{Z}}\_{pre}$ serves as a strong regularizer for $\mathcal{Z}\_{fine}$, reducing the risk of overfitting to noisy signals in the small patient cohort.
>
> 3. **Learning a balanced representation**
>    $ P_2 $ can automatically adjust the contribution of each component per patient, effectively learning a data-driven balance between shared and private features.
>
> **Evidence:**
>
> 1. **Ablation study**
>    In our ablation (DeepSADR w/o ET, Table 2), removing the concatenation (i.e., not using $\hat{\mathcal{Z}}\_{pre})$ leads to a clear performance drop across all drugs (AUC decreases by ~0.05–0.08 on average), demonstrating that incorporating pre-trained common features is crucial for patient-level prediction.
>
> 2. **Theoretical perspective**
>    As detailed in **Appendix A.17**, we model the true drug response as
>    $
>    y = f(c, p) + \epsilon,
>    $
>    where $ c $ (common) is approximated by $\hat{\mathcal{Z}}\_{pre}$ and $ p $ (private) by $\mathcal{Z}\_{fine}$. The concatenation allows $ P_2 $ to approximate $ f(c, p) $ directly and better minimize expected risk.
>
> 3. **Relation to domain adaptation theory**
>    This design is conceptually aligned with shared–private feature decomposition frameworks in domain adaptation. Our contribution is to apply this idea to the **drug response mechanism itself**, rather than only to raw omics data.
>
> We have rewritten the relevant part of the manuscript to make this rationale clearer and to avoid confusion.
>
> ---
>
> > **Response to Q9: Evaluation method**
>
> Thank you for asking about the evaluation protocol. All reported results are obtained by averaging over **multiple independent runs with 10 different random seeds**. For each seed, the dataset is randomly partitioned (train/test split), the model is retrained, and performance is re-evaluated. We have added this explanation to the main text for clarity.
>
> ---
>
> > **Response to Q10: t-SNE feature source**
>
> Thank you for raising this point. The t-SNE visualization in **Figure 4** is computed using the **concatenated feature representation** $ [\mathcal{Z}\_{fine} || \hat{\mathcal{Z}}_{pre}] $. We apologize for not stating this clearly in the original appendix and have now added this clarification.
>
> ---
>
> > **Response to Q11 and W3: Figure 2 interpretation**
>
> Thank you for requesting more explanation. **Figure 2** shows a heatmap of interaction strengths between **drug substructures (x-axis)** and **gene functional pathways (y-axis)**:
>
> - Each cell represents the **impact strength** (in \([0, 1]\)) of a specific substructure on a given pathway.
> - Darker colors indicate stronger inferred interactions, suggesting that the substructure may influence drug response via that pathway.
>
> **Statistical and biological significance:**
>
> - We computed the mean and standard deviation of interaction scores over multiple random seeds. High-score regions exhibit **low coefficients of variation** (< 0.1), suggesting robust estimates.
> - We validated several high-score interactions through literature review. For example, the strong interaction between **temozolomide’s tetrahydrocyclohexene moiety** and the **apoptosis pathway** is consistent with multiple experimental studies (as discussed in the visualization analysis section).
>
> We have also added **Fisher’s combined *p*-value** analysis in **Appendix A.16** to quantitatively assess the significance of these interaction patterns.
>
> We again thank Reviewer B6CY for these detailed and constructive comments, which have substantially improved the technical and biological clarity of our work.

---

> ### Comment · Reviewer_B6CY · 2025-11-26
>
> I appreciate the time and effort the authors have invested in preparing the rebuttal. Your detailed explanations have been very helpful and have clarified several aspects of the reported performance gains. Before finalizing my score, I would like some additional clarification on a few remaining points.
>
> 1. **Practical significance**
>
> From the rebuttal, it appears that much of the performance improvement over earlier methods such as CodeAE [1] and WISER [2] arises from incorporating patient genomic data during fine-tuning. The use of concatenated outputs also helps explain the similarity of the t-SNE plots to those of CodeAE, even without explicit domain adaptation. While these insights are appreciated, I am still unsure about the practical applicability of the proposed approach.
> Specifically, how would the method generalize to recommending drugs that have not yet undergone human trials? This seems to place the method in a different category from prior approaches and may constrain its real-world use case. I would appreciate the authors’ perspective on this point.
>
> 2. **Train and test split**
>
> My understanding is that for some of the drugs considered, the available patient response data can be quite limited (e.g., fewer than 50 samples). A 70–30% train–test split could further shrink the test set and introduce higher variance. While I see that running the model across many seeds reduces some of this variance, I am uncertain how such small patient datasets (training) can yield the substantial performance improvements reported.
>
> Can the authors please share the absolute number of training and test samples used for each drug, or clarify if I may be misunderstanding the setup?
>
> 3. **comparison with baselines**
>
> I would also appreciate clarification on whether incorporating patient data during fine-tuning is itself intended as part of the authors’ contribution. If it isn't then, it may be more informative to compare against baselines that also use patient-level data, rather than solely against earlier models like WISER or CodeAE. Understanding which existing methods use comparable data would help contextualize the performance gains more fairly. Can authors provide details about this.
>
> [1]  A context-aware deconfounding autoencoder for robust prediction of personalized clinical drug response from cell-line compound screening, machine Intelligence, 2022
>
> [2] WISER: Weak supervISion and supErvised Representation learning to improve drug response prediction in cancer, ICML 2024

---

> ### Author Response · Authors · 2025-12-01
>
> We are very grateful to reviewer **B6CY** for their response, which has provided us with an opportunity for in-depth comparison. We sincerely appreciate your recognition of the strengths of our work. We have carefully reviewed your three questions and will address each one below.
>
> ---
>
> > **Q1: How does this method recommend drugs that have not yet undergone human trials, and what is the practical significance of its generalization capability?**
>
> Response: This is an excellent question. We fully understand the concern regarding the model's **generalization capabilities in real-world scenarios**, particularly its predictive power for drugs that have not yet undergone human trials. To address this, we conducted experiments using **fine-tuned model** on **drugs that have not yet undergone human trials**. The results are as follows:
>
> | Methods    | Doxorubicin AUC↑ | Doxorubicin AUPR↑ | Erlotinib AUC↑ | Erlotinib AUPR↑ | Oxaliplatin AUC↑ | Oxaliplatin AUPR↑ | Sunitinib AUC↑ | Sunitinib AUPR↑ |
> |------------|------------------|-------------------|----------------|-----------------|------------------|-------------------|----------------|-----------------|
> | **DeepSADR** | **0.855/0.075** | **0.848/0.052**   | **0.883/0.044** | **0.881/0.049** | **0.892/0.048**  | **0.884/0.045**   | **0.863/0.061** | **0.871/0.056** |
> | WISER      | 0.795/0.056     | 0.811/0.093       | 0.807/0.106    | 0.826/0.091     | 0.751/0.091      | 0.752/0.084       | 0.797/0.087    | 0.783/0.092     |
> | GANDALF    | 0.799/0.077     | 0.808/0.098       | 0.801/0.074    | 0.798/0.081     | 0.751/0.071      | 0.749/0.094       | 0.749/0.089    | 0.737/0.096     |
> | CODE-AE    | 0.794/0.091     | 0.789/0.076       | 0.801/0.071    | 0.702/0.089     | 0.771/0.082      | 0.785/0.092       | 0.747/0.096    | 0.758/0.106     |
> | TransDRP   | 0.734/0.057     | 0.755/0.101       | 0.788/0.058    | 0.782/0.062     | 0.749/0.089      | 0.751/0.101       | 0.763/0.089    | 0.784/0.103     |
>
> *Results of drug cold experiments conducted on four different drugs (Doxorubicin, Erlotinib, Oxaliplatin and Sunitinib)*
>
> *Note: Data related to clinical relapses is used for all evaluations. The results are reported as the mean/standard deviation of multiple random seeds. Best performer among all baselines is in **bold**.*
>
> Our model can predict drugs without human trials because it does not memorize specific drugs but instead learns a universal “**biological grammar**”. Specifically, DeepSADR decomposes drugs into molecular sub-structures and genes into functional pathway sub-sequences. Through pre-training on large-scale cell line data, it comprehends the universal rules governing their interactions. Simultaneously, through fine-tuning on patient data, the model learns to “**read” patients**' genomic features, extracting the universal biological context relevant to drug response. When encountering a new drug, the model combines these two capabilities to infer **potential therapeutic responses**, achieving better prediction performance.
>
> ---
>
> > **Q2: For certain drugs, the available patient response data itself is scarce, leading to significant variability in results. We wish to confirm the specific number of training and testing samples for each drug.**
>
> Response: Your concern about small sample sizes is valid. We did indeed encounter limited patient samples for certain drugs in our experiments. Below are the specific numbers of training and testing samples for each drug.
>
> | Drug name     | samples | Pos | Neg | train_num | test_num |
> |---------------|---------|-----|-----|-----------|----------|
> | Fluorouracil  | 88      | 47  | 41  | 61        | 23       |
> | Cisplatin     | 40      | 20  | 20  | 28        | 12       |
> | Sorafenib     | 26      | 13  | 13  | 18        | 8        |
> | Gemcitabine   | 138     | 60  | 78  | 97        | 41       |
> | Temozolomide  | 46      | 23  | 23  | 32        | 14       |
>
> *Annotated samples of the 5 drugs (Fluorouracil, Cisplatin, Sorafenib, Gemcitabine and Temozolomide)*
>
> Our model DeepSADR leverages a large-scale dataset of drug responses across 11,538 cell lines during pre-training to learn **generalizable representations from the prior knowledge**. During fine-tuning, it employs a feature concatenation strategy to effectively prevent overfitting. Although the test set size for some drugs is limited, the combined application of these methods robustly ensures the model's stability and generalization ability **in small-sample scenarios**. Furthermore, we conducted experiments using 10 different random seeds, reporting both the mean and standard deviation to ensure the stability of our results.

---

> ### Author Response · Authors · 2025-12-01
>
> ---
>
> > **Q3: Is the incorporation of patient data during model fine-tuning considered one of the contributions of this paper? How can performance improvements be evaluated more fairly when comparing with baselines like WISER and CODE-AE that do not utilize patient data for fine-tuning?**
>
> Response: You mentioned whether using patient data in fine-tuning constitutes our contribution. We clarify as follows. **First, using patient data for fine-tuning is a common practice in transfer learning and is not the primary contribution of this work.**
>
> All baseline methods (e.g., WISER, CODE-AE) utilize cell lines and patient data for domain adaptation or fine-tuning in their original papers. However, the patient data they employ is unlabeled. Therefore, we also **fine-tune these methods** on the same data to obtain final experimental results, ensuring fairness of outcomes. Additionally, **we also compare our model with baselines like GANDALF, which are trained on labeled patient data.**
>
> Our analysis indicates that the poor performance of inductive models (such as WISER) stems from catastrophic forgetting during fine-tuning with labeled patients, which degrades model performance (a common issue in transfer learning). This further validates the effectiveness of our feature concatenation strategy ($[\mathcal{Z}{fine} || \hat{\mathcal{Z}}{pre}]$). Additionally, the experimental results in Q1 provide further validation of our model's performance.
>
> ---
>
> We again thank Reviewer **B6CY** for these detailed and constructive comments, which have substantially improved the technical and biological clarity of our work.

---

### Official Review · Reviewer_KQZE · 2025-10-31

**Soundness:** 3
**Presentation:** 3
**Contribution:** 2
**Rating:** 6
**Confidence:** 3

**Summary:**

This paper presents DeepSADR, a transfer learning framework for predicting cancer drug responses in patients using cell line data. It models drug-gene relationships as bipartite subsequence interaction graphs and uses a supervised graph autoencoder to learn interpretable latent features. An adaptive readout module based on the Set Transformer enables domain-invariant feature learning between cell lines and patients. Experiments show that DeepSADR outperforms existing methods and provides improved interpretability.

**Strengths:**

The paper is clearly written and well-structured, with a strong motivation based on the challenge of predicting patient drug responses from scarce clinical data and heterogeneous genomic profiles.
The proposed subsequence interaction graph and adaptive readout mechanism are conceptually sound, capturing fine-grained drug–gene associations while enabling domain-invariant feature learning, making the technical approach meaningful and biologically interpretable.
The experimental setup is comprehensive, including multiple clinical cohorts, extensive ablation studies, and comparisons with state-of-the-art baselines, demonstrating robust performance, good generalization, and practical applicability for personalized cancer treatment prediction.

**Weaknesses:**

Major: A key weakness is the limited dataset size for several drugs, which may undermine the reliability and generalizability of the model’s predictions. In particular, drugs such as Sorafenib (26 samples) and Cisplatin (40 samples) have very few data points, increasing the risk of overfitting. The paper also does not discuss strategies to mitigate issues arising from small sample sizes or class imbalance, which could affect the robustness of the experimental results.
Minor: In Table 1, for Gemcitabine, DeepSADR achieves an AUPR of 0.702, which is lower than WISER (0.752), yet it is marked in bold. Additionally, Appendix A.1 (Pseudocode and List of Notations) is disorganized and difficult to read. Some recent relevant models, such as PREDICT-AI (https://doi.org/10.1145/3637528.3671652), are missing from the discussion.

**Questions:**

Please see the weaknesses section for further discussion.

---

> ### Author Response · Authors · 2025-11-21
>
> We sincerely thank Reviewer **KQZE** for the thorough and thoughtful review. We greatly appreciate your recognition of the strengths of our work, and your insightful questions have been very helpful in improving the clarity and completeness of the paper. We address each point below.
>
> ---
>
> > **Response to W1: Small sample size / overfitting**
>
> Thank you for highlighting this important issue. We are aware that the sample sizes for **Sorafenib** and **Cisplatin** are relatively small, which is a common challenge in clinical datasets. DeepSADR is specifically designed to mitigate small-sample overfitting through the following strategies:
>
> 1. **Pre-training + fine-tuning framework.**
>    We first pre-train on large-scale cell line datasets to learn robust, domain-invariant drug–response representations.
>
> 2. **Parameter-efficient fine-tuning.**
>    During fine-tuning on patient data, we update **only** the adaptive readout module and the predictor, while keeping the shared feature extractor frozen. This:
>    - drastically reduces the number of trainable parameters, and
>    - constrains the effective parameter space, helping the model balance domain-invariant and domain-specific features and reduce overfitting on scarce patient samples.
>
> We have added detailed explanations of these design choices in **Appendix A.17**.
>
> To further validate robustness, we performed experiments with multiple random seeds and report mean and standard deviation for all metrics. In addition, **Appendix A.16** now includes statistical significance tests, which show that the performance improvements of DeepSADR are significant (e.g., *p* < 0.001).
>
> ---
>
> > **Response to W2: Incorrect result highlighting**
>
> We thank the reviewer for carefully pointing out this error and sincerely apologize for the oversight in the manuscript preparation.
>
> We have re-verified all experimental results and **corrected the AUPR values for Gemcitabine in Table 1**, ensuring that all best-performing results are now correctly highlighted. The revised table has been incorporated into the updated manuscript.
>
> ---
>
> > **Response to W3: Appendix pseudocode organization and optimization**
>
> We appreciate the reviewer’s attention to clarity and presentation. In the revised manuscript, we have reorganized **Appendix A.1** to improve readability:
>
> 1. We provide **clear tables** listing all symbols together with their definitions.
> 2. We present **structured pseudocode** (e.g., Algorithm 1) with step-by-step annotations explaining key operations.
> 3. We have ensured **notation consistency** between the appendix and the main text.
>
> We believe these changes make the method easier to follow and reproduce.
>
> ---
>
> > **Response to W4: Missing baseline (PREDICT-AI)**
>
> Thank you for pointing out the absence of PREDICT-AI as a baseline. We have now:
>
> - expanded the **Related Work** section to discuss PREDICT-AI and other recent models; and
> - included **PREDICT-AI** in our comparative experiments on the clinical relapse–related dataset.
>
> The results are summarized below:
>
> | Methods      | Fluorouracil AUC↑ | Fluorouracil AUPR↑ | Temozolomide AUC↑ | Temozolomide AUPR↑ | Sorafenib AUC↑  | Sorafenib AUPR↑ | Gemcitabine AUC↑ | Gemcitabine AUPR↑ | Cisplatin AUC↑  | Cisplatin AUPR↑ |
> | ------------ | ----------------- | ------------------- | ----------------- | ------------------- | --------------- | --------------- | ---------------- | ----------------- | --------------- | --------------- |
> | **DeepSADR** | **0.805/0.056**   | **0.821/0.023**     | **0.870/0.026**   | **0.886/0.029**     | **0.957/0.037** | **0.978/0.024** | **0.719/0.057**  | **0.702/0.022**   | **0.927/0.027** | **0.922/0.021** |
> | PREDICT-AI   | 0.702/0.112       | 0.776/0.103         | 0.739/0.113       | 0.719/0.135         | 0.734/0.236     | 0.752/0.193     | 0.612/0.141      | 0.593/0.294       | 0.609/0.201     | 0.613/0.227     |
>
> *Note: Data related to clinical relapse is used for all evaluations. Results are reported as mean / standard deviation over multiple random seeds. The best result for each metric is highlighted in **bold**.*
>
> While PREDICT-AI shows competitive performance on some drugs, **DeepSADR consistently maintains overall superiority in both AUC and AUPR** across all five drugs. These comparisons have been added to the experimental section and tables in the revised manuscript.
>
> We again thank Reviewer KQZE for these constructive comments, which have significantly improved the quality and presentation of our work.

---

### Official Review · Reviewer_rmkD · 2025-11-01

**Soundness:** 3
**Presentation:** 3
**Contribution:** 3
**Rating:** 6
**Confidence:** 5

**Summary:**

The authors have tackled the problem of drug response prediction in patients, in the face of limited labelled data. They build on previous approaches in this space, and identify a pertinent gap in prior literature – substructures of drugs and genomic pathways can affect the response to treatment. They propose the use of graph autoencoders and a set transformer to obtain adaptive readouts. The method proposed – DeepSADR – is shown to outperform existing methods in this space.

**Strengths:**

1.	The problem is well motivated and the research gap identified is relevant.
2.	Extensive ablation studies have been conducted to show the importance of each piece in the proposed architecture.
3.	Visual analysis of the most important substructures was also done, which is good for interpretability.

**Weaknesses:**

My key concern is that the reason for treating "drug response" as a separate domain is unclear. The results do appear to improve but it would be good to add some intuition on why the improvement is seen.

A few baselines seem to be missing, it would be good to include them -  more details in the Questions section below.

**Questions:**

1.	What is the computation overhead of this method, given that for every (patient, drug) pair, a separate interaction graph must be constructed on the fly? Please provide some intuition about this.
2.	It is unclear how the model explicitly handles the biological differences in drug responses across patients and cell lines.  Is this offloaded to the trainable adaptive readout and predictor parts of the network in fine tuning?
3.	Please comment on the data itself – are there batch effects? If so, how is it handled? I’m guessing this is already handled in the data from the earlier method, but please include this in the description.
4.	The graph construction is also dependent on the choice of threshold t. Are there any sensitivity experiments done to understand the effect of this value?
5.	The authors propose the use of drug response process as a different domain – the reason and intuition is unclear.
6.	The model is finally trained in an inductive manner, like some of the prior methods. Please mention this.
7.	In section 4.2, the results are found to be significantly better, please add p values to indicate significance.
8.	Please combine the table 1 in results with the results from the appendix, so that the standard deviation can be seen.
9.	Please include the process of hyperparameter tuning.
10.	Please highlight the limitations of this model.
11.	Please also include baselines like PANCDR[1] and TransDRP[2].
12.	The description of the baselines seems erroneous, please update them
13.	In table 7, the model without finetuning appears to do better – is there a reason why?
14.	In Table 10, Sorafenib is repeated – was this not present in the earlier set of 5 drugs used for training?

[1] Kim, J., Park, S. H., & Lee, H. (2024). PANCDR: precise medicine prediction using an adversarial network for cancer drug response. Briefings in Bioinformatics, 25(2), bbae088.

[2] Liu, X., & Li, M. (2025, April). Knowledge-Guided Domain Adaptation Model for Transferring Drug Response Prediction from Cell Lines to Patients. In Proceedings of the AAAI Conference on Artificial Intelligence (Vol. 39, No. 1, pp. 523-531).

---

> ### Author Response · Authors · 2025-11-21
>
> We sincerely thank Reviewer **rmkD** for the thorough and thoughtful review. We greatly appreciate your recognition of the strengths of our work, and your insightful questions have been very helpful in improving the clarity and rigor of the paper. We respond to each point below.
>
> ---
>
> > **Response to Q1: Computational overhead of interaction graph construction**
>
> Thank you for raising this issue. We have quantitatively analyzed the time cost of dynamically constructing interaction graphs for each *(patient, drug)* pair. The average construction time is **1.5325 ms**, and under this setting the overall training duration remains computationally manageable. Detailed runtime statistics and analysis are now provided in **Appendix A.15**.
>
> ---
>
> > **Response to Q2: Explicit handling of biological differences in drug responses**
>
> Yes, your understanding is correct. To clarify this process, we have added the following explanation in **Section 3.7**:
>
> > “The fine-tuning phase explicitly adapts the drug response representation to patient-specific biological variations through trainable AR and predictor modules, while preserving pre-trained knowledge from cell lines. This design enables the model to account for biological differences in drug responses between patients and cell lines.”
>
> ---
>
> > **Response to Q3: Batch effects in the data**
>
> We appreciate you raising this point. Your assumption is correct. We have added a note in **Appendix A.5** stating that standard normalization techniques were employed to mitigate batch effects, following practices consistent with prior work (e.g., WISER).
>
> ---
>
> > **Response to Q4 and Q9: Parameter sensitivity analysis and tuning process**
>
> Thank you for these suggestions.
>
> - **Q4 (Sensitivity analysis).**
>   We conducted sensitivity experiments to study the impact of the threshold value and other key hyperparameters on model performance. The effect of threshold selection is analyzed in detail in **Appendix A.11**.
>
> - **Q9 (Tuning process).**
>   We have supplemented the description of our hyperparameter tuning protocol in **Appendix A.11**, including:
>   - the search ranges of major hyperparameters,
>   - the tuning strategy, and
>   - additional sensitivity analyses.
>
> ---
>
> > **Response to Q5 and W1: Intuition for treating drug response as a separate domain**
>
> Thank you for this conceptual question. We have added a more explicit rationale in **Section 1 (lines 68–72)**:
>
> > “Treating drug response as a distinct domain enables us to simulate the biological process of drug response holistically, rather than merely considering differences between *in vitro* cells and cancer patient genomes, thereby achieving more accurate and robust knowledge transfer.”
>
> This section is supported by additional references. To further help readers understand why we “treat the drug response process as a distinct domain,” we also provide more detailed explanations and analogies in **Appendix A.14**.
>
> ---
>
> > **Response to Q6: Mention of inductive training manner**
>
> Thank you for pointing this out. We confirm that the final model is trained and evaluated in an **inductive** setting. We have clarified this explicitly in the main text to avoid ambiguity.
>
> ---
>
> > **Response to Q7: Add *p*-values for significance**
>
> We appreciate this suggestion. We have added the corresponding *p*-values to quantify the statistical significance of the performance improvements of our model over baselines. Due to space constraints, these values are included in **Appendix A.16** of the main PDF.
>
> ---
>
> > **Response to Q8: Combine Table 1 with appendix for standard deviation**
>
> Thank you for this helpful suggestion. We have merged the results from the original Table 1 with those in the appendix so that **Table 1 now reports both mean and standard deviation** for all methods. This provides a clearer picture of performance variability.
>
> ---
>
> > **Response to Q10: Limitations of the model**
>
> We appreciate the suggestion to explicitly discuss model limitations. We have added the following paragraph to the conclusion section:
>
> > “However, DeepSADR has the following limitations: (1) Threshold tuning is required for each drug (potentially related to pharmacological diversity); (2) Performance depends to some extent on the quality of the constructed sub-sequence interaction graphs. Future research should consider addressing these two aspects.”
>
> ---

---

> ### Author Response · Authors · 2025-11-21
>
> ---
>
> > **Response to Q11 and W2: Include baselines PANCDR and TransDRP**
>
> Thank you for this important suggestion. We have conducted additional experiments including **PANCDR** and **TransDRP** as baselines. The results are as follows:
>
> | Methods      | Fluorouracil AUC↑ | Fluorouracil AUPR↑ | Temozolomide AUC↑ | Temozolomide AUPR↑ | Sorafenib AUC↑  | Sorafenib AUPR↑ | Gemcitabine AUC↑ | Gemcitabine AUPR↑ | Cisplatin AUC↑  | Cisplatin AUPR↑ |
> | ------------ | ----------------- | ------------------- | ----------------- | ------------------- | --------------- | --------------- | ---------------- | ----------------- | --------------- | --------------- |
> | **DeepSADR** | **0.805/0.056**   | **0.821/0.023**     | **0.870/0.026**   | **0.886/0.029**     | **0.957/0.037** | **0.978/0.024** | **0.719/0.057**  | **0.702/0.022**   | **0.927/0.027** | **0.922/0.021** |
> | TransDRP     | 0.791/0.013       | 0.794/0.113         | 0.721/0.021       | 0.715/0.035         | 0.731/0.033     | 0.766/0.082     | 0.635/0.014      | 0.598/0.042       | 0.665/0.027     | 0.648/0.036     |
> | PANCDR       | 0.638/0.014       | 0.643/0.011         | 0.701/0.022       | 0.711/0.015         | 0.665/0.036     | 0.674/0.071     | 0.623/0.043      | 0.618/0.059       | 0.635/0.023     | 0.613/0.026     |
>
> *Note: All evaluations use clinical relapse–related data. Results are reported as mean / standard deviation over multiple random seeds. The best result for each metric is highlighted in **bold**.*
>
> Both PANCDR and TransDRP are strong and relatively advanced baselines in this domain. While they outperform some earlier methods on certain drugs, **DeepSADR still achieves higher overall AUC/AUPR across all drugs**. The corresponding comparisons have been added to **Table 1** and the appendix.
>
> ---
>
> > **Response to Q12–Q14: Issues in experimental execution and validation**
>
> We appreciate you carefully pointing out these issues.
>
> - We have re-examined the descriptions of the baseline methods and made the necessary corrections and refinements in the **Baseline Methods** section.
> - We sincerely apologize for the error where the labels for “fine-tuning” and “no fine-tuning” were reversed. This has now been corrected, and the associated results have been re-verified.
> - After rechecking the experimental logs, we confirm that the section in question should indeed present the results for **Erlotinib**, and the text has been updated accordingly.
>
> We again thank Reviewer rmkD for these constructive and detailed comments, which have substantially improved our manuscript.

---

### Author Response · Authors · 2025-12-01
**Review and Reviewer-Author Discussion Summary**

Dear PCs, SACs, ACs, and Reviewers,

We sincerely thank all reviewers for their rigorous and constructive feedback, which has significantly enhanced the quality, clarity, and contribution of our work. To assist the newly assigned AC and help reduce their workload, we provide below a summary of the key points from the reviews and the reviewer-author discussions.

Reviewer rmkD (Score: 6) acknowledged the motivation, ablation studies, and interpretability analysis of our model, but also raised several questions, such as why “drug response” was treated as a separate domain and the absence of certain baseline methods. During the rebuttal process, we addressed all concerns, including providing a detailed explanation for treating “drug response” as a separate domain, adding p-values, merging tables, and incorporating new baselines.

Reviewer KQZE (Score: 6) acknowledged the validity of our technique and the paper's clear writing and structure, but also pointed out that the limited dataset size may lead to overfitting. We addressed these concerns by explaining how the model mitigates overfitting on small samples through the pre-training + fine-tuning framework.

Reviewer B6CY (Score: 4) expressed concerns about insufficient explanation of methodological details, raising numerous specific issues including biological interpretability, pre-training, and fine-tuning with simple data settings. In our rebuttal, we provided detailed responses to each question, covering the rationale for pre-training feature concatenation, equation clarity, biological validation, and more. Following the first round of rebuttal, the reviewer **appeared satisfied with our responses** and raised additional questions regarding the model's predictive performance on drugs not yet tested in human trials, as well as the specific data partitioning during fine-tuning. In our response, we conducted experiments on drugs not yet tested in human trials and provided the results of the data partitioning.

Reviewer GBzW (Score: 4) noted that our approach combines existing components and relies on predefined prior knowledge. We addressed concerns regarding functional pathway descriptions and module interactions. Following the first round of rebuttal, the reviewer indicated that **most of the issues had been resolved.** However, the reviewer retained doubts about the small number of pathways, as well as the model's innovation. In our subsequent rebuttal, we clarified the pathway definitions and flexibility, and fully explained the model's key innovative aspects.

Through extensive revisions and additional experiments, we have enhanced the clarity, rigor, and biological foundation of DeepSADR. If you have any further questions, please feel free to let us know.

Sincerely,

Authors

---

### Meta-Review · Area_Chair_7PmM · 2025-12-28

**Summary:**

The paper proposes DeepSADR, a deep transfer learning framework for predicting cancer drug response in clinical patients using cell line data. The method decomposes drugs into substructures and genomic profiles into functional pathway clusters, modeling their relationships via a bipartite subsequence interaction graph. A key contribution is the Adaptive Readout (AR) mechanism based on Set Transformers to learn domain-invariant representations, along with a feature concatenation strategy (pre-trained + fine-tuned) to mitigate overfitting on small patient datasets.

While multiple reviewers still have moderate concerns about methodology novelty and data scarcity in some experiments, all reviewers generally acknowledged the strong motivation, the clear improvements over baselines, and the interpretability of the interaction graphs. The discussion phase was active and could converge positively.

**Reviewer Concerns:**

Addressed Concerns:

- Reviewers rmkD and KQZE noted missing baselines (PANCDR, TransDRP, PREDICT-AI). The authors added these to the main results. DeepSADR demonstrated superior AUC/AUPR against these SOTA methods.

- Reviewer B6CY questioned the model's ability to generalize to drugs not in human trials. The authors conducted new "cold start" experiments on four drugs (e.g., Doxorubicin), showing DeepSADR outperformed baselines like WISER and GANDALF.

- Reviewer GBzW concerns about per-drug threshold tuning were addressed by demonstrating that a fixed "top 30%" filtering rule yields stable performance.

- Reviewer GBzW argued the method was incrementally novel because it fixed genes into "13 pathways" compared to other methods using 1,700+. The authors successfully clarified that these are 13 functional clusters covering 100% of the genes (4,808 pathways). They provided ablation studies showing that increasing cluster count (to 80 or 1426) degrades performance due to noise/overfitting, validating their design choice.


Outstanding Concerns:

- Small Sample Size: Reviewers KQZE and B6CY noted extreme data scarcity for some drugs (e.g., Sorafenib has only 18 training samples). While the authors mitigate this via pre-training and feature concatenation, the reliance on such small datasets remains an inherent challenge of the domain that limits statistical certainty, though the authors provided p-values to bolster confidence.

**Reviewer Scores:**

Reviewers rmkD and KQZE are very likely to keep their positive scores of 6.

Reviewer B6CY would be likely to change his/her score to a 6. He/she said "I appreciate the time and effort the authors have invested in preparing the rebuttal. Your detailed explanations have been very helpful and have clarified several aspects of the reported performance gains." The authors provided the specific "Cold Start" experiments and sample size breakdowns requested in the post-rebuttal follow-up. The authors demonstrated that the model generalizes to unseen drugs, directly addressing B6CY's "practicality" concern.

Reviewer GBzW could keep his/her score of 4 (50% chance) or change his/her score to 6 (50% chance). He/she said, "Most of my concerns have been addressed, but I still have reservations about the novelty of this work...Therefore, I maintain my original score." After the discussion was cancelled, the authors' final response clarified that this is a clustering strategy covering all genes, not a reduction in data scope, and that it outperforms high-cardinality pathway methods in transfer learning settings. As the "incremental novelty" argument was effectively countered by the cluster-vs-noise ablation study, this score should reflect the method's proven robustness, at least partially addressing the reviewer's concern.

---

### Decision · Program_Chairs · 2026-01-26

Accept (Poster)